https://doi.org/10.1038/s42003-022-04165-3　　**OPEN**
# Neurocomputational mechanisms of affected beliefs

Laura Müller-Pinzler [1✉], Nora Czekalla [1], Annalina V. Mayer[1], Alexander Schröder [1], David S. Stolz [1], Frieder M. Paulus [1] & Sören Krach [1]

The feedback people receive on their behavior shapes the process of belief formation and self-efficacy in mastering a particular task. However, the neural and computational mechanisms of how the subjective value of self-efficacy beliefs, and the corresponding affect, influence the learning process remain unclear. We investigated these mechanisms during self-efficacy belief formation using fMRI, pupillometry, and computational modeling, and by analyzing individual differences in affective experience. Biases in the formation of self-efficacy beliefs were associated with affect, pupil dilation, and neural activity within the anterior insula, amygdala, ventral tegmental area/ substantia nigra, and mPFC. Specifically, neural and pupil responses mapped the valence of the prediction errors in correspondence with individuals' experienced affective states and learning biases during self-efficacy belief formation. Together with the functional connectivity dynamics of the anterior insula within this network, our results provide evidence for neural and computational mechanisms of how we arrive at affected beliefs.

[1] Department of Psychiatry and Psychotherapy, Social Neuroscience Lab, University of Lübeck, Ratzeburger Allee 160, D-23538 Lübeck, Germany.
✉email: laura.muellerpinzler@uni-luebeck.de

Self-efficacy can be defined as a person's subjective conviction that he/she can overcome challenging situations through his/her own actions[1]. To successfully perform goal-directed actions, humans must learn from incoming information, thereby forming beliefs about the world and about themselves enmeshed in this world. According to economic theory, learning should result in accurate beliefs that represent an internal model of the world that is suitable to inform decision making. Novel theoretical frameworks, among others by Bromberg-Martin and Sharot[2], emphasize that besides the instrumentality (i.e. accuracy) of beliefs, they may also carry intrinsic value in and of themselves, thus shaping the learning process and how people ultimately arrive at their beliefs[3]. In this regard, affective states, such as happiness about one's own good health prognosis, represent intrinsic values that individuals are inclined to optimize during belief formation[2,4]. To demonstrate this entanglement of affect and belief formation, we applied a learning task that induces affective reactions during the process of forming conceptually novel beliefs about one's abilities to master a task[5,6]. Specifically, we focused on the primary affective states elicited by self-efficacy beliefs – the self-conscious emotions of embarrassment and pride – and their impact on the beliefs. By exerting experimental control over failures and successes during the process of self-efficacy belief formation, we were able to assess how experienced affect relates to computational mechanisms of belief formation and the underlying activity of neural systems, linking neural and physiological mechanisms with shifts in preferences for information of positive or negative valence during learning.

Affective states are considered to guide cognitive processing, representing embodied and experiential information about the positive or negative value of what people encounter[7,8]. It is proposed that this internal affective information is integrated with external information to shape beliefs that rather than being objective, are motivated and biased by subjective feelings about the beliefs themselves, leading to a recursive influence of beliefs and affective states on each other[2,9,10]. Previous studies supported aspects of Bromberg-Martin and Sharot's framework[2] by demonstrating that internal beliefs and external feedback can elicit emotions like happiness, pride, or embarrassment[11–17]. Affective states also have been shown to alter decision making[13,18,19] and cognitive processes like situational judgments or learning styles[7]. Social anxiety, low self-esteem, or depression, which are likely associated with more negative affective reactions to self-efficacy beliefs, have also been found to bias social learning[5,20–22]. These findings provide support for the overall rationale of the formation of affected beliefs, that is, the notion that beliefs are fundamentally shaped by motivational biases as well as affective experiences during feedback processing. However, the question remains open of which neurophysiological mechanisms can explain how emotions elicited during learning are associated with biases in belief formation.

Neuroscientific studies provided initial evidence that common brain areas map the value of stimuli, actions, and their motivational relevance during social and non-social learning and decision making[23,24]. Prediction errors, that is, the mismatch of prior expectation and a situation's outcome, are minimized by updating beliefs during learning. These are generally processed in the dopaminergically innervated ventral striatum, but also in the orbitofrontal cortex or the amygdala during learning[24–27]. However, more recent findings suggest that there are distinct and unique neural computations which potentially reflect the impact of the prominent motivational and emotional processes during belief formation. For example, studies have shown that distinct value-related neural processes in subregions of the anterior cingulate cortex (ACC) are recruited depending on whether information about oneself or another agent is processed[28,29]. Other findings revealed that activation in the ventral striatum was modulated when the social context changed from a private to a public situation, suggesting that the presence or absence of other people influenced the sensitivity to the reward value of certain decisions[30]. Biases specific to self-related learning, which are absent when one is learning about another person[5,31], have been associated with differences in the tracking of negative prediction errors[32]. In this regard, the ventromedial prefrontal cortex (vmPFC) shows valence-specific encoding of self-related feedback, which has been shown to predict an optimism bias in belief updating[31,33].

Affective states triggered after personal failures or successes are particularly important when people acquire novel self-concepts[34] and develop an initial understanding of themselves as being self-efficacious individuals in a novel task environment. Central to the entanglement of affect and such self-efficacy beliefs is the assumption that people are highly motivated to perform well and maintain or even construct a positively shaped self-image[35,36]. Within this process, performance feedback elicits self-conscious emotions, such as pride in the case of success[13,37,38], but also embarrassment if one fails to achieve the expected outcome[14,37,39]. Self-conscious emotions differ from other emotional concepts as they essentially involve self-referential evaluations and the activation of self-concepts[37]. Thus, when it comes to emotional experiences in the context of a performance situation, pride or embarrassment are theoretically more valid constructs to capture differences in affective experiences than e.g., the basic emotion happiness. In the past, it has been demonstrated that these self-conscious emotions are not only a consequence of the situation but also directly affect behavior. Pride experiences function as a motivator to persevere[38]. In contrast, embarrassment experiences rather lead individuals to stop their current behavior, withdraw, and appease others[40,41]. For the process of belief formation, it has been argued that specifically the dorsomedial frontal cortex (dMFC), the ventral and dorsal anterior insula (vAI/ dAI), and the amygdala, brain areas involved in action monitoring as well as emotional processing, integrate affective states with outcome information[42]. Therefore, the anterior insula (AI) has been regarded, among other brain regions, as an integrative hub for motivated cognition and emotional behavior[42,43]. Similarly, dopaminergic midbrain nuclei in the ventral tegmental area and substantia nigra (VTA/ SN) are associated with attention processes, and at the same time, with events (i.e. reward cues) that are of motivational relevance specifically during learning[44,45].

While current frameworks support the idea that intrinsic outcomes such as affective states may impact the formation of self-efficacy beliefs[2,4], studies on this issue have not yet probed this framework as a whole. We aim to bridge this gap by showing how emotional states relate to biases in the formation of self-efficacy beliefs, and how they are associated with preferences for information of positive or negative valence. For this purpose, we tested the effects of individual differences in the affective reactions during learning. Using trial-by-trial updates of performance expectations in a conceptually novel task environment, we computed prediction error learning rates by fitting computational learning models revealing valence-specific learning biases. As predicted by current frameworks, individual differences in the experience of the emotions embarrassment and pride were distinctly related to biases in the formation of self-efficacy beliefs. Biased learning and affect were jointly related to the neural processing of valence-specific prediction errors in the AI, amygdala, VTA/ SN and mPFC as well as pupillary reactivity in favor of the preferentially used information to update the belief. Increases in valence-specific functional connectivity of the dAI with the amygdala, VTA/ SN and mPFC support the notion of an

integrative mechanism of affective and motivational processes within the dAI[46,47]. These findings provide insights into brain networks involved in computational biases associated with emotional experiences, and coherently support current theoretical frameworks integrating affective experiences in the process of belief formation.

## Results

**Measuring self-efficacy belief formation**. In the present experiment, $n = 39$ subjects (26 females, aged 18–28 years; $M = 22.3$; $SD = 2.65$) completed the task in the MRI. Another $n = 30$ subjects (24 females, aged 18–32 years; $M = 23.3$; $SD = 3.97$) completed the task outside the MRI as a behavioral study. During the MRI scanning, eye-tracking data was additionally obtained in all but three subjects (see section "Methods" for more details). To examine the formation of self-efficacy beliefs we used the Learning Of Own Performance (LOOP) task[5,6].

In brief, in the LOOP task participants are asked to estimate specific characteristics of properties (e.g., heights of buildings, weights of animals, numbers of things, or distances between objects). By manipulating the performance feedback, participants are led to form novel beliefs on their own and the other person's cognitive estimation abilities. In the fMRI sample, participants perform the LOOP task in the MRI scanner while a confederate (presented as another participant) ostensibly performs the task simultaneously in an adjacent room. After each trial, participants receive a manipulated performance feedback for the last estimation (see Fig. 1a). During the entire experiment, participants take turns in performing the estimation task themselves (Self condition) or observing the other participant performing the task (Other condition). Before each trial, participants are asked to rate either their own or the other person's expected performance for the upcoming trial, enabling us to examine the process of self- and other-related belief formation. The design of the LOOP task provides a High Ability and a Low Ability condition, resulting in overall four feedback conditions: Agent condition (Self vs. Other) x Ability condition (High Ability vs. Low Ability; see Fig. 1b and the "Methods" section for a detailed description of the task). In previous studies, we showed that over time, participants adjusted their expected performance ratings according to the feedback, allowing for an assessment of valence-specific self- and other-related learning processes[5,6].

**Selection of computational models for self-efficacy belief formation**. Following a model-free behavioral analysis (see Supplementary Note 1), we modeled the participants' behavior by means of learning rates. Changes in expectations were modeled through updates from prediction errors (PEs) to test different learning rates for PEs with positive vs. negative valence and Self vs. Other (Supplementary Figs. 1, 2). In line with our previous studies, the winning model was a Valence Model, including separate learning rates for positive and negative PEs for Self vs. Other (Model 8; for a more detailed description of this model and the whole model space, see "Methods" section). This model received the highest sum PSIS-LOO score (approximate leave-one-out cross-validation (LOO) using Pareto smoothed importance sampling (PSIS)[48]), out of all models (for all PSIS-LOO scores see Supplementary Table 1). In addition, Bayesian model selection (BMS) resulted in a protected exceedance probability of $pxp > .999$ for this model and a Bayesian Omnibus Risk of $BOR < .001$. The expected model frequency was 46.53. Thus, the extended Valence Model was selected for all further analyses of learning parameters, allowing for a comparison of valence-specific learning rates. The time courses of performance expectation ratings predicted by our

winning model successfully captured trial-by-trial changes in the actual expectations due to PE updates within each of the ability conditions at the individual subject level ($R^2 = 0.46 \pm 0.28$ [$M \pm SD$]), supporting the validity of the model in describing the subjects' learning behavior. In addition to revealing PE valence-specific learning, which could not be directly assessed via model-free behavioral analyses, posterior predictive checks also confirmed that the winning model captured the core effects in our model-free analysis (see Supplementary Note 2, Fig. 1c, Supplementary Table 2; for parameter correlations see Supplementary Table 3). Exploratory analyses with learning rates from Model 5 showed that our results were unaffected by the $w$ parameter modulating learning from more extreme feedback in the winning Model 8 (see Supplementary Note 3).

**Replication of the negativity bias for the formation of self-efficacy beliefs**. Participants showed higher learning rates when forming self-efficacy beliefs than when forming beliefs about the other person's performance (main effect of Agent: $F_{(1,67)} = 5.77$, $p = .019$, $\eta^2 = 0.017$, partial $\eta^2 = 0.079$). There was also a main effect of Prediction Error Sign ($F_{(1,67)} = 5.22$, $p = .025$, $\eta^2 = 0.011$, partial $\eta^2 = 0.072$; categorical comparison of learning rates for positive vs. negative PEs) and a significant interaction of Agent x Prediction Error Sign ($F_{(1,67)} = 21.47$, $p < .001$, $\eta^2 = 0.040$, partial $\eta^2 = 0.243$), which replicates earlier findings of a bias towards more negative updating during self-efficacy belief formation ($t_{(68)} = -3.53$, $p < .001$, $d = -0.425$, $M\alpha_{Self/PE+} = 0.25$, $SD = 0.13$; $M\alpha_{Self/PE-} = 0.35$, $SD = 0.20$)[5]. Forming beliefs about another person's performance did not reveal a significant bias towards more negative updating ($t_{(68)} = 2.67$, $p = .009$, $d = 0.321$; $M\alpha_{Other/PE+} = 0.27$, $SD = 0.16$; $M\alpha_{Other/PE-} = 0.24$, $SD = 0.15$; see Fig. 1d). There was no significant main effect or interaction for Group ($p > .097$).

**Associations of self-efficacy belief formation with affective experience**. We hypothesized that self-efficacy belief formation is associated with affective experience. In line with Bromberg-Martin and Sharot[2] we expected that individuals with more negative affective experience would update their self-efficacy beliefs in a more negative way. To quantify associations between learning behavior and affect, individual differences in the overall experience of embarrassment and pride during the task were used as between-subject measures. Embarrassment and pride ratings were only weakly correlated ($\rho_{(68)} = -.10$, $p = .436$), indicating that the experience of embarrassment and pride during the task represent two rather independent affective components with respect to the self-related feedback (see Supplementary Table 4 for a more detailed correlation table). The bias in the formation of self-efficacy beliefs (Valence Learning Bias = $(\alpha_{Self/PE+} - \alpha_{Self/PE-})/(\alpha_{Self/PE+} + \alpha_{Self/PE-})$[5,49,50] was negatively linked to the reported experience of embarrassment during the task ($\rho_{(68)} = -.24$, $p = .043$), that is, more negative updating behavior was associated with increased embarrassment ratings. In contrast, the Valence Learning Bias was positively linked to the emotion of pride ($\rho_{(68)} = .55$, $p < .001$). A regression predicting the Valence Learning Bias with both affect ratings simultaneously revealed independent effects of pride ($\beta = 0.56$, $t_{(66)} = 5.81$, $p < .001$) and embarrassment ($\beta = -0.22$, $t_{(66)} = -2.30$, $p = .025$; $R^2 = .41$, $F_{(1,66)} = 22.90$, $p < .001$, $f^2 = 0.64$). When controlling for differences in the feedback participants received before rating their affective experience, correlations between emotions and Valence Learning Bias do not significantly change and the overall pattern of associations remains consistent. This indicates that the experience of self-conscious emotions during successful and unsuccessful performances was tied to the way in which people

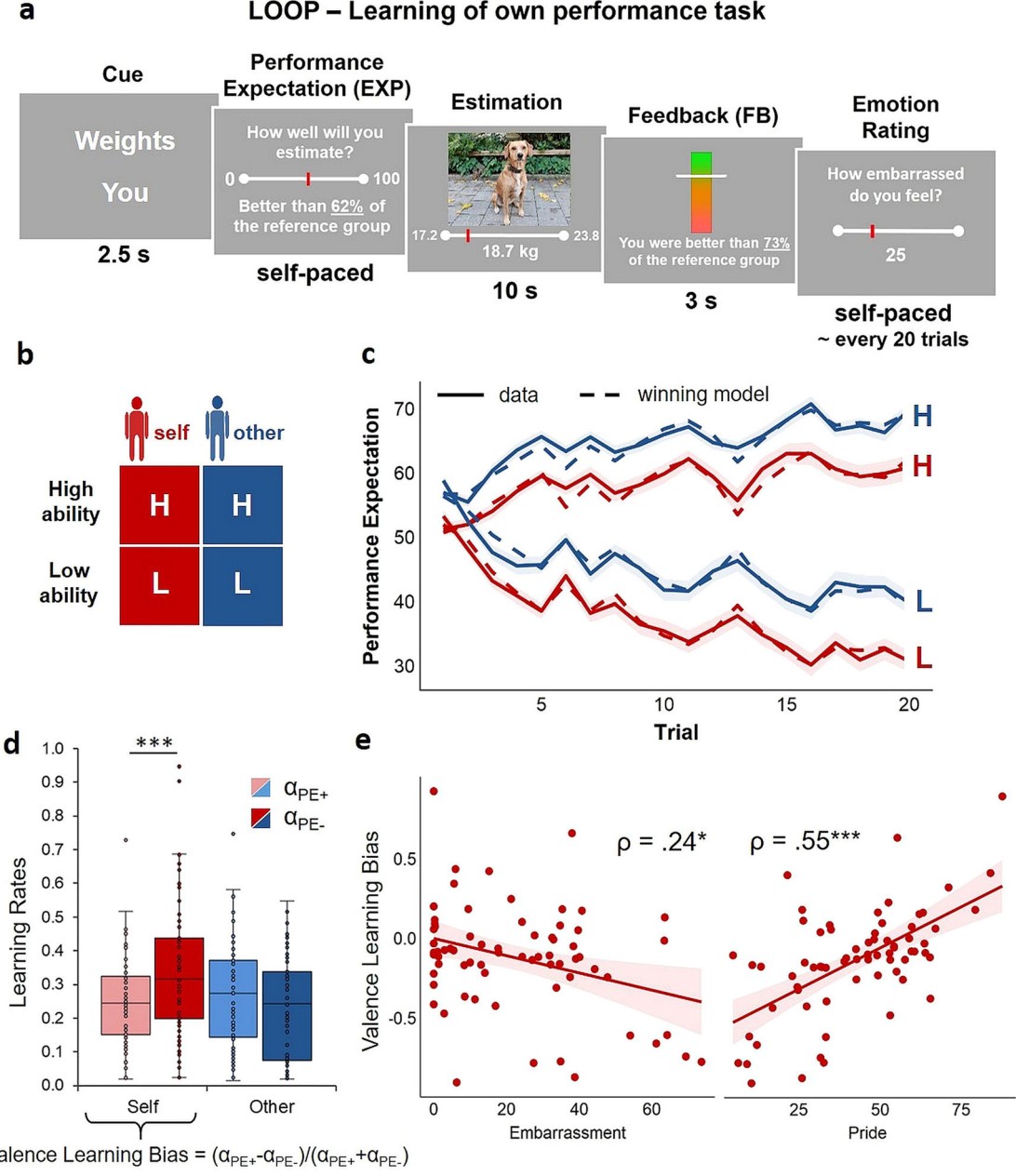

**Fig. 1 Trial sequence and timing, experimental conditions, modeling of learning behavior, learning rates and their association with self-conscious emotions. a** Shows a stylized version of the estimation task. A cue at the beginning of each trial indicated the following estimation category and the agent whose turn it was. After providing their performance expectation ratings (EXP), participants were asked an estimation question, followed by the corresponding performance feedback. After approximately every 20 trials, participants were asked to rate their current emotional state (pride, embarrassment, happiness, stress/ arousal). **b** The LOOP task contained two experimental factors, Ability level (High Ability vs. Low Ability) and Agent (Self vs. Other), resulting in four feedback learning conditions that can be distinguished by different estimation question types (e.g. estimation of weights). **c** Predicted and actual performance expectation ratings across time. The behavioral data indicate that participants adapted their performance expectation ratings (solid lines) to the provided feedback, thus learning about the allegedly distinct performance levels. The winning valence-specific learning model captured the participants' behavior, as indicated by a close match of actual performance expectations with the predicted data (dashed lines). Shaded areas represent the standard errors for the actual performance expectations. **d** Learning rates derived from the winning Valence Model indicate that there was a bias towards increased updating in response to negative prediction errors ($\alpha_{PE-}$) in contrast to positive prediction errors ($\alpha_{PE+}$) for the formation of self-efficacy beliefs. Colored bars indicate the first and third quartile of the data, the line marks the median. Whiskers extend from the upper (lower) box borders to the largest (smallest) data point at most 1.5 times the interquartile range above (below) the respective border. Data with more extreme values than this are displayed as individual points; ***$p < 0.001$, indicates a significant negativity bias during the formation of self-efficacy beliefs. **e** Correlation plots and Spearman correlations of self-related Valence Learning Bias and embarrassment as well as pride experience during the experiment. *$p < 0.05$, ***$p < 0.001$.

updated their self-efficacy beliefs (see Fig. 1e). Furthermore, the way in which participants processed the performance feedback in order to update their self-efficacy beliefs was associated with their self-esteem. Specifically, participants with higher self-esteem showed more positive updating, $\rho_{(68)} = .33$, $p = .006$ (fMRI sub-sample: $\rho_{(38)} = .35$, $p = .030$), which strengthens the assumption that prior beliefs about the self have a direct impact on how individuals learn novel information about new abilities[5,51].

**Pupil dilation slopes are associated with surprise and valence of prediction errors, in line with a negative learning bias.** Previous research has successfully linked changes in pupil diameter to surprise, PEs and learning[52,53] as well as emotional experiences and arousal[14,54]. Thus, we hypothesized that PE tracking is linked to changes in pupil diameter. To corroborate our assumption that changes in pupil diameter, as indicated by the slope of the change in pupil size during the processing of self-related feedback, reflect increased arousal or attention in association with greater PEs, we regressed trial-by-trial variability in the pupil slope on PE surprise (continuous effect of unsigned PEs) and PE valence (continuous effect of signed PEs; see Fig. 2a)[55]. The linear mixed model revealed a significant positive effect for PE surprise ($\beta = 0.067$, $t_{(325.9)} = 2.16$, $p = .032$, 95% $CI = [0.006; 0.127]$) and a significant negative effect for PE valence ($\beta = -0.113$, $t_{(30.7)} = -2.52$, $p = .017$, 95% $CI = [-0.200; -0.025]$; see Supplementary Fig. 3). First, we observed an effect of PE surprise, insofar as the more surprising the feedback was with respect to trial-by-trial prior expectations, the more the pupil dilated. Second, the results indicate that pupil dilation was greater with decreasing PE values, thus linking negative PEs, rather than positive PEs, to greater dilation (i.e. effect of PE valence). Potentially, these PE valence effects indicate increased arousal and attention towards more negative PEs, in line with the negativity bias that we found in learning rates.

**Pupil dilation response to prediction error valence is associated with affect and learning bias.** It has been suggested that pupil dilation reflects differences not only between stimuli but similarly between individual biases during decision making (see Fig. 2b for examples of individual differences)[56]. We thus expected individual differences in self-efficacy belief formation and affective experience to be associated with differences in pupil responses to PEs. To test this assumption, we introduced individual differences in learning and self-conscious emotions as between-subject covariates into the linear mixed models assessing trial-by-trial pupil slopes. These analyses demonstrated that individuals who experienced more embarrassment showed stronger pupil dilations scaling with more negative PEs, while pupil slopes did not correlate with PEs in individuals with lower embarrassment (significant interaction of embarrassment and PE valence: ($\beta = -0.0004$, $t_{(32.5)} = -2.59$, $p = 0.015$, 95% $CI = [-0.0006; -0.0001]$; no main effect for embarrassment: $\beta = 0.002$, $t_{(34.2)} = 0.42$, $p = 0.679$, 95% $CI = [-0.009; 0.014]$; see Fig. 2c). These effects were reversed when pride ratings, instead of embarrassment ratings, were included in the model (interaction pride and PE valence ($\beta = 0.0005$, $t_{(34)} = 3.18$, $p = 0.003$, 95% $CI = [0.0002; 0.0007]$; main effect of pride: $\beta = -0.00006$, $t_{(34.1)} = -0.01$, $p = 0.991$, 95% $CI = [-0.01132; 0.01120]$). The Valence Learning Bias modulated the relationship between PE valence and pupil slopes in the same way (interaction Valence Learning Bias and PE valence ($\beta = 0.38$, $t_{(31.8)} = 3.02$, $p = 0.005$, 95% $CI = [0.13; 0.62]$; main effect of Valence Learning Bias: $\beta = 0.33$, $t_{(50.9)} = 1.04$, $p = 0.308$, 95% $CI = [-0.30; 0.97]$), indicating that participants with a more negative Valence Learning Bias showed a negative correlation of pupil dilation and PEs,

whereas participants with no bias or a positive bias showed less differentiation in pupil dilation in response to the valence of the PE.

**Common neural activations associated with PE surprise and distinct activations for PE valence.** On the level of the brain, we assessed the association of PE tracking with neural activity and tested whether there is a specific response pattern with respect to self- and other-related belief formation. To do so, we computed the effects of continuous trial-by-trial PE surprise and PE valence as parametric weights to assess neural aspects of learning more specifically (see Fig. 3a). Increased PE surprise was associated with greater activation of the mPFC for Self and Other as well as clusters in the left insula/ temporal pole/ frontal orbital gyrus (bilaterally for Other; see Fig. 3c and Supplementary Table 5). There was no significant difference between Self and Other (p < .001 uncorrected), indicating that there is no evidence for distinct neural processes of error tracking between agents.

The assessment of PE valence revealed a distinct pattern for self- and other-related belief formation: Self-related PE valence was positively associated with increased activation of the NAcc/ VS, mPFC, bilateral angular gyrus/ superior parietal lobule/ lateral occipital gyrus and precentral gyrus, showing stronger activation scaling with more positive PEs (Fig. 3b and Supplementary Table 6). There was no effect for other-related PE valence, and a direct comparison of self- vs. other-related PE valence effects revealed stronger associations in the NAcc/VS for Self (right: x, y, z: 12, 17, −4, $t_{(38)} = 5.23$; k = 2; left: x, y, z: −9, 26, −1, $t_{(38)} = 5.77$, k = 19). This supports the assumption that the valence of the feedback has a specific value when feedback refers to the self as compared to another person. Although behavioral data and learning rates clearly emphasize the greater importance of negative over positive PEs, there were no significant negative associations with PE valence in the neural data ($p < .001$ uncorrected). Additional analyses assessing differences between the feedback conditions for Agent and Prediction Error Sign are presented in the Supplementary Information (Supplementary Note 4, Supplementary Fig. 4, Supplementary Data 1, and Supplementary Table 7).

**Neural activity in response to self-related PE valence is associated with affect, learning bias, and pupil dilation.** To assess how biases in learning as well as affective experience and pupil dilation were associated with valence-specific PE processing on the single trial level, multiple general linear models (GLMs) were performed. These included the Valence Learning Bias, self-conscious emotions, and a score representing a valence bias for pupil dilation responses to positive vs. negative PEs (Pupil Dilation Bias = PupilSlope$_{Self/PE+}$ - PupilSlope$_{Self/PE-}$) as between-subject covariates for PE valence tracking. Analyses within our predefined regions of interest (ROIs) revealed that the more negative the Valence Learning Bias was, the more neural activity increased with more negative PEs in the bilateral dAI, vAI, amygdala, mPFC, and VTA/ SN (all results are $p < 0.05$ family-wise error (FWE) corrected at peak level within ROIs; see Fig. 3d, e, Supplementary Data 2). In other words, the more positively participants learned about themselves (i.e., more positive Valence Learning Bias), the more neural activity increased with more positive PEs in these regions (see Fig. 3d, e, and Supplementary Fig. 5). Overall, higher experience of embarrassment showed similar associations with stronger activity with more negative PEs in the right dAI, bilateral amygdala, and VTA/ SN. Trend effects for embarrassment were found in the left dAI, bilateral vAI, and mPFC. In line with this, lower experience of pride showed the same association in the dAI and vAI,

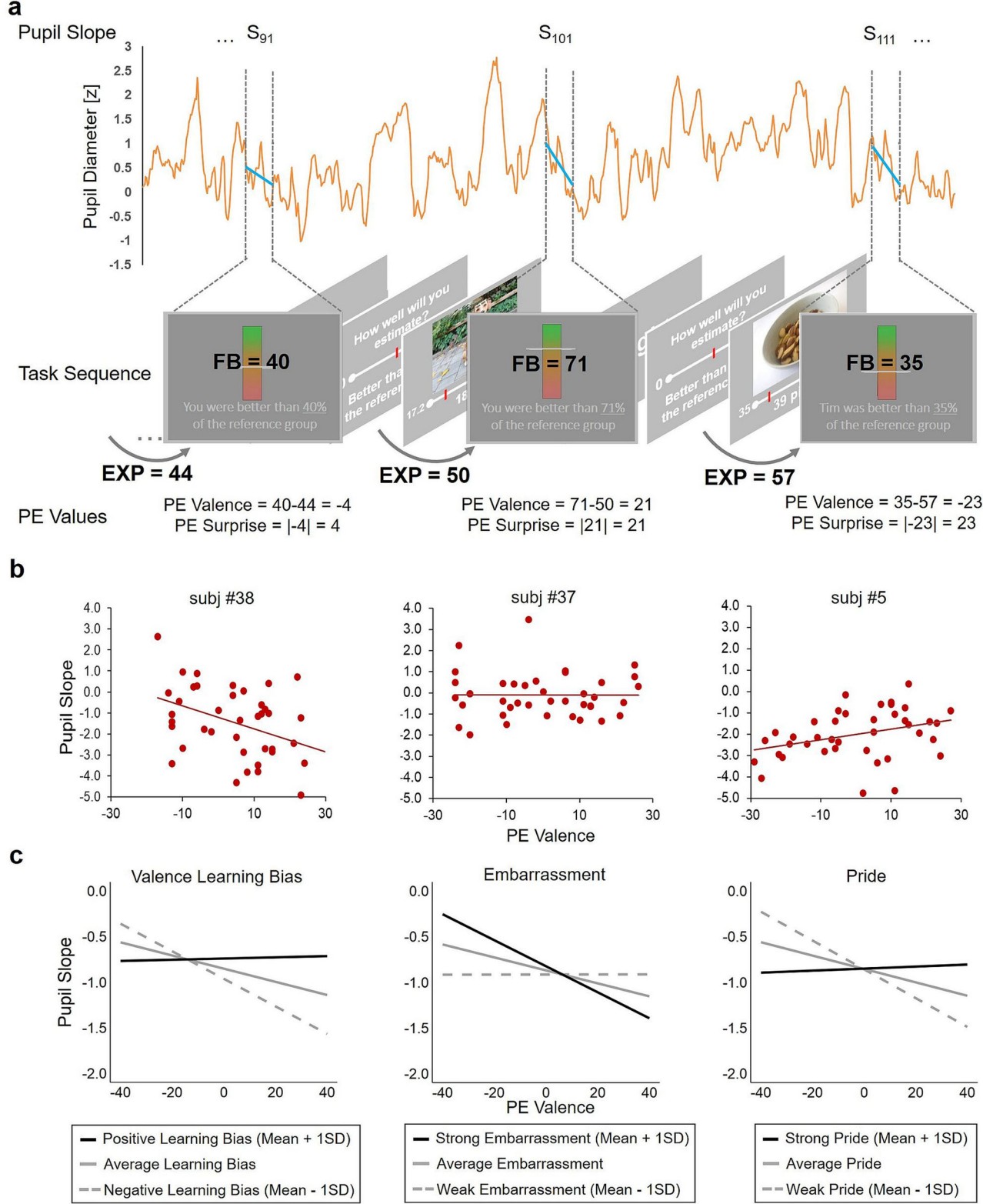

**Fig. 2 Association of pupil slopes with prediction error (PE) valence and individual pupil response differences explained by differences in Valence Learning Bias, embarrassment and pride experience. a** Example of pupil diameter trace over three trials for one subject (orange line) and trial-specific fitted linear slopes (blue lines) for the feedback phase of each trial. PE values are calculated with the participant's current performance expectation (EXP) and the following feedback value (FB), and PE valence represents the signed PE while PE surprise represents the unsigned PE. **b** Three exemplary scatter plots show the association of pupil slopes with PE valence and illustrate the variance between subjects. Trend lines are fitted by linear regression. **c** Illustration of the impact of the three between-subject covariates, Valence Learning Bias (left), embarrassment (middle) and pride experience (right) explaining differences in the associations of PE valence and pupil slope. The plots show the data as predicted by the multi-level models for the mean covariate (grey line) and the mean covariate +/− 1 standard deviation (SD; black line and gray dashed line).

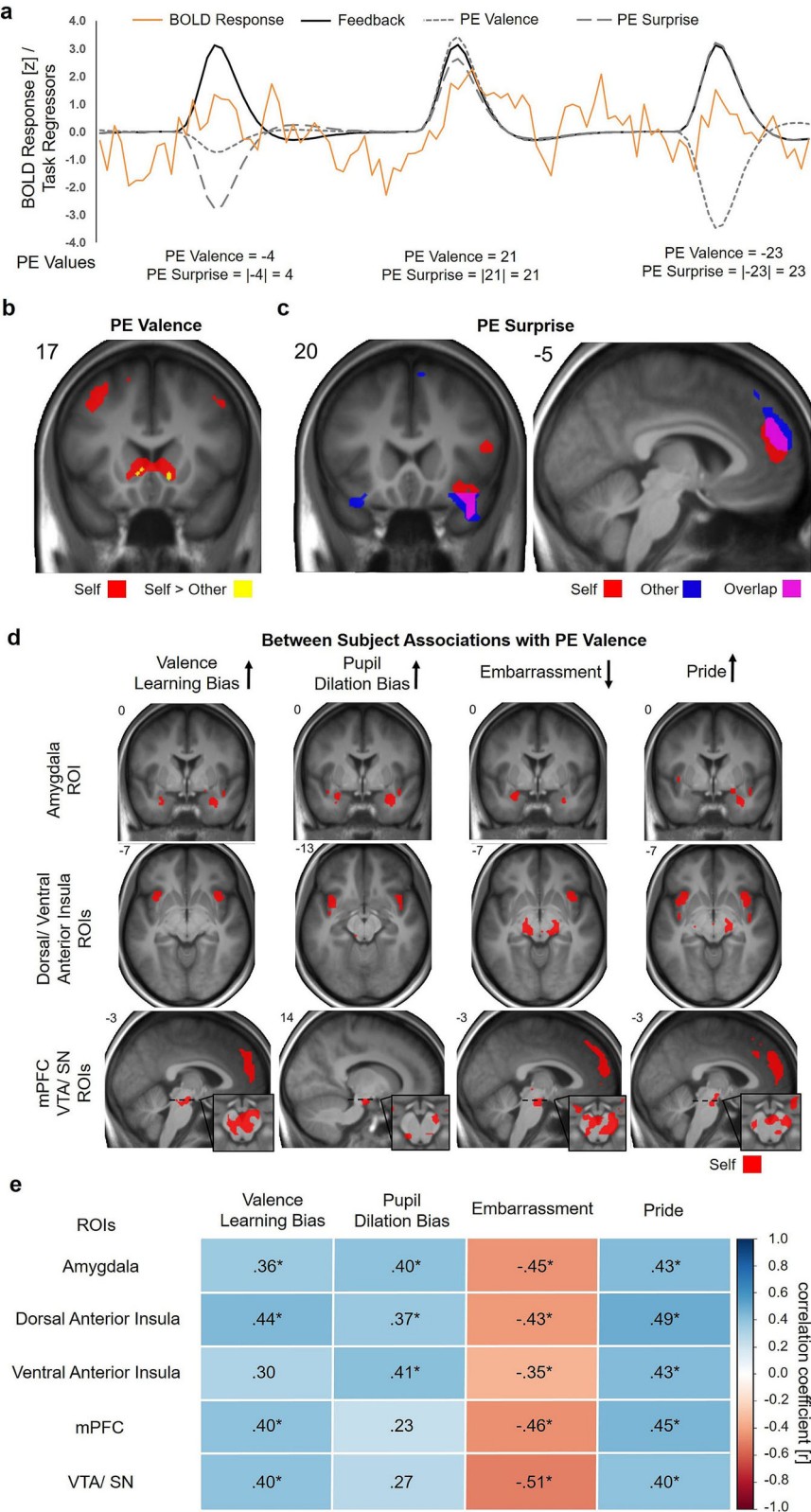

amygdala, VTA/ SN and mPFC. Additional analyses revealed that effects for embarrassment and pride were mainly independent (see Supplementary Note 5). Similarly, the more negative the Pupil Dilation Bias was, the stronger the activation of the dAI and vAI, amygdala and VTA/ SN with more negative PEs. Thus, the greater the response of this neural system for more negative PEs, the greater was the preference for negative information during learning

as well as the negativity of the affective experience. This gained multi-modal support by similar associations of the Valence Learning Bias and affect with the pupil dilation response, which reflects the activity of this underlying neural system. In contrast, participants who showed a greater response of this neural system to positive PEs also had a preference for positive information during learning and reported more positive affect.

**Fig. 3 Common neural activations associated with prediction error (PE) surprise, distinct activations for self-related PE valence and individual response differences to PE valence explained by differences in Valence Learning Bias, embarrassment and pride experience, and pupil dilation.**
**a** Exemplary BOLD response over three trials for one subject (orange line) and regressors for the feedback phase of each trial (black line; the originally separate regressors for self- and other-related feedback are combined here for display purposes). PE valence (small dashed) and PE surprise (large dashed) are added as parametric modulators in addition to the feedback regressors. PE values are calculated as shown in Fig. 2. **b** PE valence was associated with increased activation of the NAcc/VS, mPFC, bilateral angular gyrus/ superior parietal lobule/ lateral occipital gyrus and precentral gyrus when participants formed self-efficacy beliefs (Self). **c** PE surprise was associated with activation of the mPFC and the bilateral insula/ temporal pole/ frontal orbital gyrus during the formation of self- and other related beliefs (uncorrected $p < 0.001$ for display purposes; see Supplementary Table 5 for FWE corrected statistics). **d** Neural tracking of PE valence during the formation of self-efficacy beliefs was modulated by between-subject variables. Black arrows indicate the direction in which the covariates are coded in the analyses. Clusters refer to $p < 0.005$, uncorrected for display purposes; see Supplementary Data 2 for FWE corrected statistics. **e** Pearson correlations for parameter estimates derived from the whole areas of our predefined ROIs with the Valence Learning Bias, Pupil Dilation Bias, embarrassment and pride are color-coded. *$p < 0.05$, FDR corrected.

**Functional connectivity of the dorsal anterior insula depends on prediction error valence in line with the negativity bias.** Due to the dense anatomical and functional connections between the dAI and (para)limbic as well as frontal brain regions[46,47] we tested whether dAI connectivity increased with more negative PEs. To do so, we assessed functional connectivity dynamics of the left and right dAI, as these were activated during feedback processing for self- and other-related feedback, independent of Agent and PE valence. Using psychophysiological interaction (PPI) analyses we calculated the interaction of the continuous PE valence and the time series extracted from the left and right dAI seed regions separately for Self and Other on the first level. The two agents were contrasted against each other on the second-level GLM, as we were specifically interested in connectivity dynamics that might reflect the differential learning from negative PEs when processing self-relevant information. Contrasting the PPI effects for PE valence between Self and Other demonstrated that during the formation of self-efficacy beliefs, functional connectivity dynamics of the right dAI with the bilateral amygdala, mPFC and VTA/ SN ($p < 0.05$, FWE corrected at peak level within ROIs) more strongly aligned with the negativity of the PEs. The left dAI showed a weaker but similar spatial distribution, with significant differences between self- and other-related PE valence for the left amygdala and VTA/ SN ($p < 0.05$, FWE-corrected, see Fig. 4a, b and Supplementary Table 8). Thus, those brain regions that preferentially tracked PEs of negative valence in individuals with increased negative affect and learning biases also showed connectivity dynamics with the dAI in a similar direction during self-efficacy belief formation. Individuals who showed more pronounced differences in functional connectivity, that is, stronger functional connectivity for negative PEs during Self>Other, also showed a more negative Valence Learning Bias, although this pattern was not fully consistent across all ROIs (see Fig. 4c).

## Discussion
Belief formation is essentially biased, and various studies have shown how it is shaped by motivations[3,36,57,58]. Here, we extend these findings and show that the affect, which people experience during learning, also is linked to belief formation and its underlying neural processes. Our computational modeling results imply that biases in the formation of self-efficacy beliefs in mastering a conceptually novel task are associated with the experience of the self-conscious emotions of embarrassment and pride. Critically, on the level of neural systems, the valence of prediction errors (PEs) is associated with biases in self-efficacy belief formation and the negativity of the affective experience. Individual differences in the response preference for negative PEs, as indicated by the pupil dilation response and activation of the AI, amygdala, mPFC, and VTA/ SN, are associated with a more

negative learning bias and negative affective experience, hinting at a neurobiological system that integrates affect during learning.

The novel framework on the value of beliefs proposed by Bromberg-Martin and Sharot[2] nicely details how beliefs elicit emotions, while at the same time, these emotions shape how beliefs are updated in a reciprocal relationship. Based on this framework as well as previous research on self-conscious emotions, a negative belief about one's abilities, i.e. a negative self-efficacy belief, should elicit stronger embarrassment after failures and reduced pride after successes[14,37]. According to the present data, the association of the learning bias with affect supports this notion, as individuals who experienced more negative (embarrassment) and less positive affect (pride) when receiving self-related feedback were also inclined to update their self-efficacy beliefs in a more negative way. At the same time, negative emotions guide the information processing at various stages, including perception, attention, and decision-making, as discussed in the context of motivated cognition[4]. This reciprocal relationship finally results in a biased formation of self-efficacy beliefs and in self-efficacy beliefs that are both drivers of affect and influenced by emotional responses to incoming information. Here, embarrassment is a particularly relevant example illustrating this recursive relationship: The fear of failure, as is often discussed in the context of social anxiety (disorder)[5,14,20,59], leads to shifts in expectations and attention (threat monitoring) towards negative information. At best, this results in reparative behavior and performance improvement[60,61], and at worst, it leads to a vicious cycle of fear and pathologically increasing negative beliefs about the self[62]. This is reflected in the present findings, when individuals who experienced more intense embarrassment ended up with lower self-efficacy beliefs.

Emotions shape learning processes in different ways. First, emotions can influence how information is processed in the brain by adaptively shifting attention towards salient aspects of the situation[63,64]. Second, emotions entail arousal, which intensifies internal rehearsal and evaluations, leading to increased learning[7,8,63], although these processes often interact and are intricately related[4]. The increased pupil dilation in response to negative PEs in our study is in line with both increased salience of and attentional shifts towards negative PEs[52,53] or increased arousal elicited by negative PEs[14,54]. In this regard, we believe that the stronger impact of positive or negative information on pupil responses and brain reactivity maps arousal and affect according to the valence of individual learning biases and affective experiences.

On the neural level, the anterior insula (AI), specifically, has been suggested to function as an integrative hub for motivated cognition and emotional behavior[42,43]. While ventral aspects of the AI are associated with affective processing, emotions, and physiological arousal[43,47,65–67], dorsal aspects of the AI are strongly associated with the detection of salient events, allocation

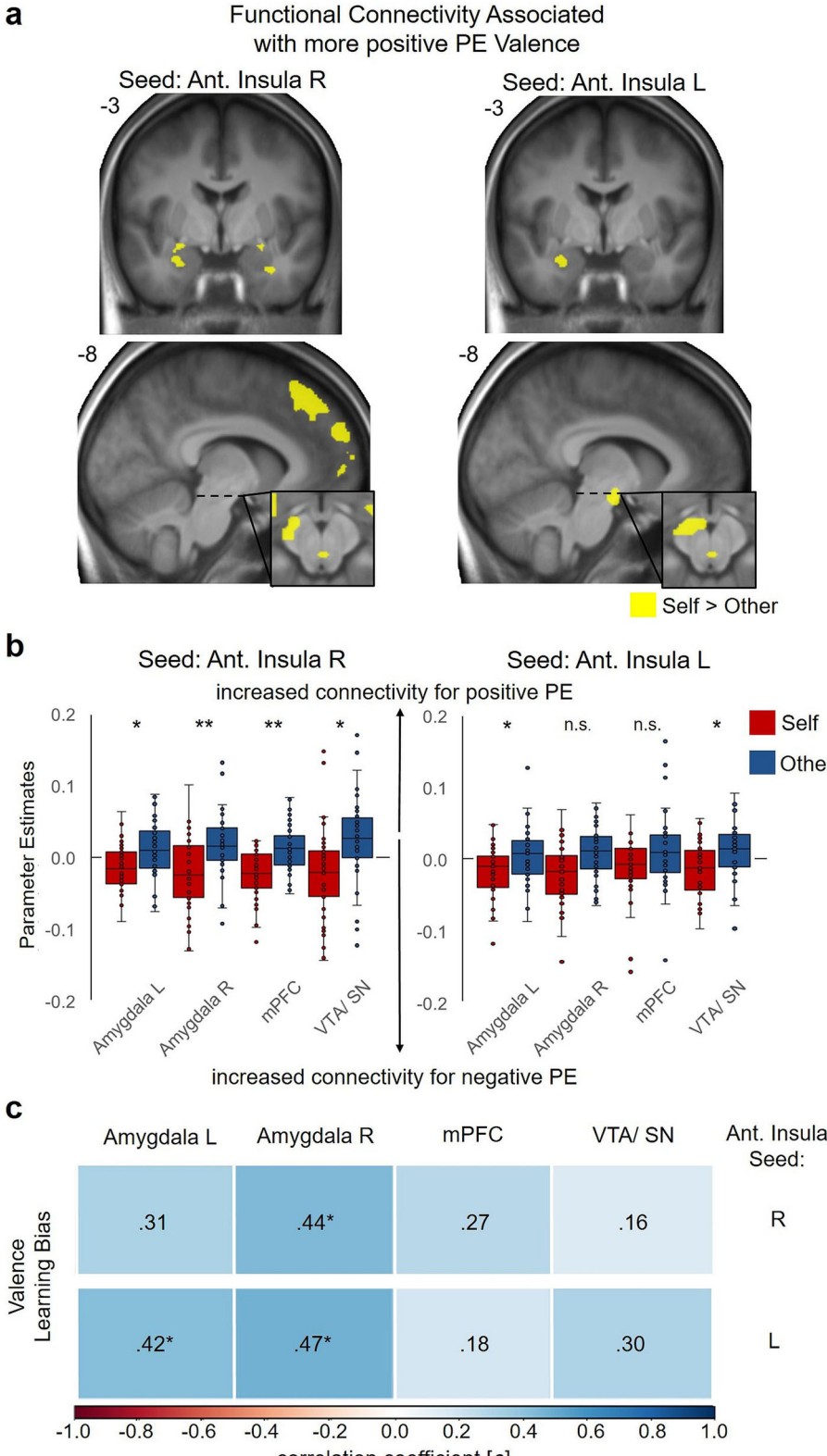

of attentional resources, executive working memory[68,69] and also surprise PEs and uncertainty during learning[70–72]. These findings suggest that the functions of the AI provide a physiological basis for how emotions are translated into biased, motivated, or affected beliefs[42,43]. A similar role, as a link for the attention-emotion interaction, has also been suggested for the amygdala[42,64], which showed similar responses in our task. The

functional connectivity dynamics of the dAI, matching the modeled learning rates with a stronger impact of self-related negative PEs, underline the insula's role as an integrative hub[73] that receives and forwards signals affecting information processing in other brain regions[46,47].

Tracking of PEs in the dopaminergically innervated VTA/ SN is influenced by motivational factors during learning[44]. The

**Fig. 4 Differences in functional connectivity of the dorsal anterior insula during prediction error (PE) valence tracking during the formation of self-and other-related beliefs and associations with the Valence Learning Bias. a** Increased functional connectivity of the dorsal anterior insula for the negative effect of PE valence in the predefined ROIs (amygdala, mPFC, VTA/ SN; $p < 0.005$ uncorrected for display purposes; contrast Self vs. Other). **b** Functional connectivity dynamics of the dorsal anterior insula plotted separately for the formation of self- and other-related beliefs. For display purposes, parameter estimates are plotted separately for Self and Other and refer to the peak voxels of the contrast Self vs. Other that are reported in Supplementary Table 8. Colored bars indicate the first and third quartile of the data, the line marks the median. Whiskers extend from the upper (lower) box borders to the largest (smallest) data point at most 1.5 times the interquartile range above (below) the respective border. Data with more extreme values than this are displayed as individual points; *$p < 0.05$, **$p < 0.01$. **c** Spearman correlations of the Valence Learning Bias with the functional connectivity dynamics between the dorsal anterior insula (seed region reported on the right side) and the amygdala, mPFC and VTA/ SN associated with PE valence for self- vs. other-related learning are color-coded. *$p < 0.05$, FDR corrected.

subjective value of self-related information varies strongly between subjects, as indicated by response patterns of the VTA/ SN to gains or losses[74]. In this line, we believe that the present results reflect individual response tendencies at a very basic level of PE tracking. On higher layers of the computational hierarchy, regions in the ACC and mPFC are also associated with PE tracking and value representation[28,75,76] and have been previously associated with biases in learning[31,33,77]. Affect and arousal could therefore bias learning on various stages of the computational hierarchy of PE processing, from more basic dopaminergic midbrain responses to more abstract value representations in the neocortex[78]. While the directionality of the effects remains to be determined, the dynamics in the functional connectivity of the dAI suggest a modulatory role in this process. Here, information is forwarded to and/ or integrated from the VTA/ SN and mPFC, the same regions whose response to the valence of PEs was also modulated by differences in learning bias and affective experience. This strengthens the idea that the AI may play a role in shifting responses to negative or positive information in other brain regions (e.g. by shifting attention and by affective tagging) or that it already may receive stronger signals in response to PEs of negative or positive valence from midbrain regions and the mPFC. The tracking of the absolute error, PE surprise[55], independently of the agent, suggests that there is a common and valence-independent coding of surprise in the insula and the mPFC that could be sufficient to complete the learning task per se. As our results indicate, however, the valence of the PE is relevant for understanding the trajectories of how individuals form self-efficacy beliefs. This is implicated by a valence-dependent, additive shift in the error-related BOLD response of these regions that corresponds to individual differences in the learning bias and affective experience. As a result, besides the main effect of surprise on the BOLD response of the AI and the mPFC, individuals who form more negatively biased self-efficacy beliefs and experience more negative affect, also have greater error-related responses in the case of negative PEs in contrast to positive PEs. This congruency in the modulation of the U-shaped surprise function hints at a neurocomputational mechanism of how affect may shape the formation of beliefs, as proposed previously[2]. Some of the key findings of the present study emerged at the level of individual differences. We observed a wide inter-individual variance in the affective experience during the task and in the learning bias, that is, the type of information participants preferentially used to update self-efficacy beliefs. While, on average, we found a negativity bias during the formation of self-efficacy beliefs, just under a third of the participants still showed a positive learning bias, underlining the importance of individual factors and the meaningfulness of variability. Studies suggest not only that biases in belief formation differ between tasks[3,5,79] but also that they depend on contextual factors like stress[6,80]. An individual's ability to adjust his/her current information processing strategy to the context might be adaptive[2]: for example, adaptation to an increased relevance of

negative or threat-related information during stress[80] or coping with a negative self-concept following social stress by means of more self-beneficial belief updating[6]. It might also be adaptive for people who fear negative feedback to pay more attention to failure-related information in order to learn and circumvent potential future failures[57]. However, it is not always straightforward to determine under which conditions a strategy is adaptive or whether the affective experience can ameliorate the individual's well-being. A maladaptive consequence of biased self-efficacy beliefs becomes apparent in psychiatric disorders such as depression and social anxiety, in which amplified negative updating can lead to persistently distorted self-views and overly negative beliefs about one's own capabilities in everyday life[20,21,81–83].

Emotions experienced during learning are linked to computational mechanisms and manifest in distributed neural activity during belief formation. In particular, neural activity of the AI, amygdala, VTA/SN, and mPFC and pupil responses map the valence of PEs in correspondence with the experienced affect and the learning bias that people show during belief formation. The more negative balancing in the functional connectivity dynamics of the dAI during the processing of self-related PEs within this network outline a scaffold for neural and computational mechanisms integrating affect during belief formation. The results of our empirical implementation of the framework on the value of beliefs[2] have broader implications concerning any context that provides personal evaluations based on behavioral performance. Here, the focus on the affective experience during learning provides a deeper understanding of how feedback manifests in self-efficacy beliefs, which may in turn have a relevant impact on developmental processes and future behavior.

## Methods

**Participants.** The study was approved by the ethics committee of the University of Lübeck (AZ 18–066), was conducted in compliance with the ethical guidelines of the American Psychological Association (APA), and all subjects gave written informed consent. Participants were recruited at the University Campus of Lübeck, were fluent in German, and had normal or corrected-to-normal vision. In the MRI 39 participants (26 females, aged 18–28 years; $M = 22.3$; $SD = 2.65$) completed the study. We initially recruited 48 participants, but had to exclude six participants who did not believe the cover story of the task and three participants who did not attentively complete the task until the end (e.g. participants reported that they were too tired or the ratings indicated that they stopped responding to the estimation task). During the MRI scanning, eye-tracking data was additionally obtained and could be analyzed in all but three subjects who had insufficient data quality (resulting in $n = 36$ for pupil data analyses). We recruited an additional 30 participants (24 females, aged 18–32 years; $M = 23.3$; $SD = 3.97$), who completed the study as a behavioral study outside the MRI to increase the sample size for computational modeling results (resulting in an overall $N = 69$ for behavioral data analyses). For more details on the sample characteristics, see Supplementary Table 9.

**Learning of own performance task.** The learning of own performance (LOOP) task enables participants to incrementally learn about their own or another person's alleged ability in estimating properties. The task was previously introduced and validated in a set of behavioral studies[5]. For the LOOP task, all participants were invited to take part in an experiment on cognitive estimation together with a

confederate, who was allegedly another participant. In contrast to the fMRI study, for the behavioral study, two participants were invited and tested together instead of introducing a confederate. Participants were informed that they would take turns with the other participant/ confederate, either performing the task themselves (Self) or observing the other person performing (Other). Participants were asked to estimate different properties (e.g. the height of houses or the weight of animals) On a trial-by-trial basis, participants received manipulated performance feedback in two distinct estimation categories for their own estimation performance and for the other person's estimation performance. Unbeknownst to the participant, one of the two categories was arbitrarily paired with rather positive feedback while the other was paired with rather negative feedback (e.g. height of houses = High Ability condition and weight of animals = Low Ability condition or vice versa; estimation categories were counterbalanced between Ability conditions and Agent [Self vs. Other] conditions). This resulted in four feedback conditions with 20 trials each (Agent condition [Self vs. Other] x Ability condition [High Ability vs. Low Ability]). Trials of all conditions were intermixed in a fixed order with a maximum of two consecutive trials of the same condition. Performance feedback was provided after every estimation trial, indicating the participant's own or the other person's current estimation accuracy as percentiles compared to an alleged reference group of 350 university students who, according to the cover story, had been tested beforehand (e.g. "You are better than 94% of the reference participants."; see Fig. 1a). The feedback was defined by a sequence of fixed PEs with respect to the participants' current belief about their abilities. The current belief was calculated as the average of the last five performance expectation ratings per category, which started at 50% before participants actually rated their performance expectation. This procedure led to varying feedback sequences between participants but kept PEs mostly independent of the participants' performance expectations and ensured a relatively equal distribution of negative and positive PEs across conditions (Self: mean positive PE = 13.6, SD = 1.8 (mean frequency = 20.3); mean negative PE = −12.6, SD = 1.4 (mean frequency = 19.7); Other: mean positive PE = 13.0, SD = 1.3 (mean frequency = 19); mean negative PE = −13.1, SD = 1.1 (mean frequency = 21)). At the beginning of each trial, a cue was presented indicating the estimation category (e.g. height) and the agent whose turn it was (e.g. you or Tim). Afterwards participants were asked to state their expected performance for this trial on a scale with the same percentiles used for feedback. In order to increase motivation and encourage honest response behavior, participants were informed as part of the cover story that accurate expected performance ratings would be rewarded with up to 6 cents per trial, that is, the better the match between their expected performance rating and their actual feedback percentile, the more money they would receive. Following each performance expectation rating, the estimation question was presented for 10 s. During the estimation period, continuous response scales below the pictures determined a range of plausible answers for each question. Participants indicated their responses by navigating a pointer on the response scale with an MRI-compatible computer mouse. Subsequently, feedback was presented for 3 s (see Fig. 1a). Jittered inter-stimulus-intervals were presented following the cue (mean: 4 * TR (0.992 s), range: 2–6 * TR), estimation (mean: 4.5 * TR, range: 2.5–6.5 * TR) and feedback phase (mean: 6 * TR, range: 4–8 * TR) for the fMRI task with jitters distributed in a uniform distribution with steps of 0.5 * TR. All stimuli were presented using MATLAB Release 2015b (The MathWorks, Inc.) and the Psychophysics Toolbox[84]. The fMRI task was completed in two separate 20-min sessions with a short break in between.

Before starting the experiment, all participants answered several questions about their self-efficacy beliefs and completed a self-esteem personality questionnaire (Self-Description Questionnaire-III (SDQ-III)[85]). During the LOOP task, participants were also asked to rate their current levels of embarrassment, pride, happiness and stress/ arousal on a continuous scale ranging from not at all (coded as 0) to very strong (coded as 100). During the whole task four emotion rating phases, including all four emotions, were presented, each following a trial of one of the four experimental conditions (e.g. Self - High Ability). The two emotion rating phases following self-related feedback were averaged to obtain a rating for the experience of self-conscious affect (embarrassment and pride) during the formation of self-efficacy beliefs. Following the task, participants completed an interview including ratings about self-efficacy beliefs, were debriefed about the cover story, and reimbursed for their time before leaving. The whole procedure took approximately 2 h.

## Statistics and reproducibility

*Behavioral data analysis and modeling.* To illustrate effects in our behavioral data, a model-free analysis was performed on the participants' expected performance ratings for each trial. We conducted a linear mixed model (LMM) fitted with restricted maximum likelihood (REML) including the Ability condition (High Ability vs. Low Ability) x Agent condition (Self vs. Other) as factorial and Trial (20 Trials) as continuous predictors. Intercept, Ability condition, Agent condition, and Trial were modeled as fixed and random effects (see Supplementary Note 1 for results).

Following model free analyses, dynamic changes in self-efficacy beliefs, that is, performance expectation ratings, were then modeled using PE delta-rule update equations (adapted Rescorla-Wagner model[86]). For the learning models the following PE delta-rule update equation was used (EXP = Performance expectation rating, FB = feedback, PE = prediction error, $\alpha$ = learning rate):

$$\text{EXP}_{t+1} = \text{EXP}_t + \alpha\,\text{PE}_t; \text{while } \text{PE}_t = \text{FB}_t - \text{EXP}_t \tag{1}$$

The model space contained three main models, which varied with regard to their assumptions about biased updating behavior when forming self-efficacy beliefs (see Supplementary Fig. 1). The simplest learning model used one single learning rate for all conditions for each participant, thus not assuming any learning biases (Unity Model). The second model, the Valence Model, included separate learning rates for positive PEs ($\alpha_{\text{PE+}}$) vs. negative PEs ($\alpha_{\text{PE-}}$) across both ability conditions, thus suggesting that the valence (positive vs. negative) of the PE biases self-efficacy belief formation. The third model, the Ability Model, contained a separate learning rate for each of the ability conditions, indicating context-specific learning. In addition, learning rates were either estimated separately for Self vs. Other or across Agent conditions. The Valence Model with separate learning rates for Self vs. Other (Model 5), which was the winning model in our previous studies[5,6], was further extended by adding a weighting factor that reduced learning rates towards the ends of the feedback scale (percentiles close to 0% or 100%), under the assumption that participants would perceive extreme feedback values to be less likely than more average feedback[87]. In the first of these models (Model 7), a linear decrease of the learning rates was assumed, beginning at 50% and ending at 0% and 100%. A weighting factor $w$ was fitted for each participant, defining how strongly the linear decrease was present for each individual. Since many of the variables people encounter in everyday life (e.g., many test results) approximately follow a normal distribution with extreme values being less likely, for the second model of this kind (Model 8), we assigned the relative probability density of the normal distribution to each feedback percentile value. Again, a weighting factor $w$ was fitted for each individual, indicating how strongly the relative probability density reduced the learning rates for feedback further away from the mean. The initial beliefs about the own and the other participant's performance ($\text{EXP}_1$) were estimated as free parameters separately for Self and Other as well as both Ability conditions, resulting in four additional model parameters. The linear (LD) and normal decay (ND; values depicted in Supplementary Fig. 2) weighted by the weighting factor $w$ that reduced the learning rates towards the ends of the scale were introduced in the learning models in the following way:

$$\text{EXP}_{t+1} = \text{EXP}_t + \alpha\,\text{PE}_t(1 - w\,\text{LD}); \text{for the linear decrease;} \tag{2}$$

$$\text{EXP}_{t+1} = \text{EXP}_t + \alpha\,\text{PE}_t(1 - w\,\text{ND}); \text{for the normal decrease.} \tag{3}$$

In contrast to our previous studies in which we implemented the LOOP task with fixed feedback sequences, here, feedback depended on the participants' current expectations and thus differed between participants and conditions. Reduced learning rates towards the ends of the feedback scale, which may systematically confound learning rates between participants and conditions, were thus accounted for in Models 7 and 8 (see Supplementary Information; Supplementary Fig. 2). To test whether the participants' performance expectation ratings can be better explained in terms of PE learning as compared to stable assumptions in each Ability condition, we included a simple Mean Model, with a mean value for each task condition (Model 9).

*Model fitting.* For model fitting, we used the RStan package (Stan Development Team, 2016. RStan: the R interface to Stan. R package version 2.14.1.), which implements Markov chain Monte Carlo (MCMC) sampling algorithms. All of the learning models in the model space were fitted for each participant individually, and posterior parameter distributions were sampled for each participant. A total of 2400 samples were drawn after 1000 burn-in samples (overall 3400 samples; thinned with a factor of 3) in three MCMC chains. We assessed whether MCMC chains converged to the target distributions by inspecting; $\hat{R}$ values for all model parameters[88]. Effective sample sizes ($n_{\text{eff}}$) of model parameters, which are estimates of the effective number of independent draws from the posterior distribution, were typically greater than 1500 (for most parameters and subjects). Posterior distributions for all parameters for each of the participants were summarized by their mean as the central tendency, resulting in a single parameter value per participant that we used in order to calculate group statistics.

*Bayesian model selection and family inference.* For model selection, we estimated pointwise out-of-sample prediction accuracy for all fitted models separately for each participant by approximating leave-one-out cross-validation (LOO; corresponding to leave-one-trial-out per subject;[48,89]). To do so, we applied Pareto smoothed importance sampling (PSIS) using the log-likelihood calculated from the posterior simulations of the parameter values as implemented by Vehtari et al.[48]. Sum PSIS-LOO scores for each model as well as information about $\hat{k}$ values – the estimated shape parameters of the generalized Pareto distribution – indicating the reliability of the PSIS-LOO estimate are depicted in Supplementary Table 1. As summarized in Supplementary Table 1, very few trials resulted in insufficient parameter values for $\hat{k}$ and thus potentially unreliable PSIS-LOO scores (on average 1.1 trials per subject with $\hat{k} > 0.7$ for the winning model[48]). BMS on PSIS-LOO scores was performed on the group level, accounting for group heterogeneity in the model that best describes learning behavior[90]. This procedure provides the protected exceedance probability for each model (*pxp*), indicating how likely a given model is to have a higher probability of explaining the data than all other models in

the comparison set. The Bayesian omnibus risk (*BOR*) indicates the posterior probability that model frequencies for all models are all equal to each other[90]. We also provide difference scores of PSIS-LOO in contrast to the model that won the BMS, which can be interpreted as a simple 'fixed-effect' model comparison (see Supplementary Table 1[48,89]). Model comparisons according to PSIS-LOO difference scores were qualitatively comparable to the BMS analyses for our data. Posterior predictive checks were conducted following model selection by quantifying whether the predicted data could capture the variance in performance expectation ratings for each subject within each of the experimental conditions using linear regression analyses. Additionally, to assess whether the winning model captured the core effects in the behavioral data, we repeated the model-free analysis, which we had conducted on the behavioral data, with the data predicted by the winning model (see Supplementary Note 2 for results).

*Statistical analyses of learning parameters.* Model parameters, i.e. learning rates, of the winning models for all experiments were analyzed on the group level. A repeated measures ANOVA was calculated on the learning rates with the factor Agent (Self [$\alpha_{Self/PE+}$, $\alpha_{Self/PE-}$] vs. Other [$\alpha_{Other/PE+}$, $\alpha_{Other/PE-}$]) and factor Prediction Error Sign (PE + [$\alpha_{Self/PE+}$, $\alpha_{Other/PE+}$] vs. PE- [$\alpha_{Self/PE-}$, $\alpha_{Other/PE-}$]; in line with the winning model, the term bias corresponds to the categorical distinction between feedback with positive PEs vs negative PEs) as well as Group as a between-subject factor (fMRI vs. behavior), testing whether the formation of self-efficacy beliefs was more valence-specific than forming beliefs about another person's performance.

To associate learning biases with self-conscious affect, that is, embarrassment and pride, as well as self-esteem (SDQ-III subscale scores), we calculated a normalized learning rate valence bias score for self-related learning (Valence Learning Bias = ($\alpha_{Self/PE+}$ - $\alpha_{Self/PE-}$)/($\alpha_{Self/PE+}$ + $\alpha_{Self/PE-}$)[5,49,50]. Spearman correlations were calculated between Valence Learning Bias, affect ratings, and self-esteem scores. Statistical tests were performed two-sided if not mentioned otherwise. All statistical analyses on the behavioral data apart from the modeling procedure were performed using *jamovi* (Version 1.2.27, The jamovi project (2020), retrieved from https://www.jamovi.org).

*Pupil data analysis.* For the fMRI sample, eye-tracking data were assessed during scanning. Pupil diameter and gaze behavior were recorded non-invasively in one eye at 500 Hz using an MRI-compatible Eyelink-1000 plus device (SR Research, Kanata, ON, Canada) with manufacturer-recommended settings for calibration and blink detection. Stimuli were presented on a TFT display (32", Active area 698.4 mm(H) x 392.85 mm(V); Pixels 1920 × 1080; NordicNeuroLab's LCD monitor [NNL MRI InroomViewingDevice; NordicNeuroLab AS, Møllendalsveien 65 C, 5009 Bergen, Norway]), located 50 cm from the observer, in an otherwise dark room. The task has been optimized with respect to eye-tracking by controlling the global luminance of the stimuli as well as the local luminance of the feedback scale. Due to insufficient pupillometry data quality, three participants had to be excluded from the analyses (final sample *n* = 36). Pupil data were preprocessed by cutting out periods of blinks, and values in this gap were interpolated by piecewise cubic interpolation. The pupil trace was subsequently z-normalized over the whole session. To characterize the pupil dilation for each trial by a single value, we calculated a linear slope for each feedback phase of three seconds. Summarizing the pupil dynamics during a single trial with a linear slope is a robust and valid measure for an arousal related pupil response to stimuli of comparable lengths[14,91,92] building up after 1–2 s until reaching a plateau after more than approximately 6 s[54,93]. Pupil traces were only analyzed for the Self condition as there was an offset in the pupil diameter at the beginning of the feedback presentation (see Supplementary Fig. 6). The strong difference in the pupil diameter between the Agent conditions is expected given the greater arousal[94] after the cognitive effort when estimating properties. While the pupil slope is a robust measure of rather sustained relative change during stimulus presentation, this offset in the diameter at the beginning of the feedback presentation makes it impossible to draw valid comparisons of these slopes between the Agent conditions as greater pupil diameter will result in greater negative slopes compared to smaller pupil diameter. The linear mixed models (LMMs) with pupil slopes as dependent variable were fitted including intercept, PE valence, and PE surprise both as fixed and random effects. In three separate models either embarrassment ratings, pride ratings, or the Valence Learning Bias (Covariates) were included as second-level covariates as well as their interaction with PE valence (see Supplementary Note 6, Supplementary Figs. 7–9 for supporting analyses on the linear mixed models). The model description of the full model was as following:

$$s_{i,j} = \gamma_{0,0} + \gamma_{1,0} * PE\ Valence_{i,j} + \gamma_{2,0} * PE\ Surprise_{i,j} + \gamma_{0,1} * Covariate_j$$
$$+ \gamma_{1,1} * Covariate_j * PE\ Valence_{i,j} + v_{0,j} + v_{1,j} * PE\ Valence_{i,j} \qquad (4)$$
$$+ v_{2,j} * PE\ Surprise_{i,j} + \varepsilon_{i,j}$$

*fMRI data acquisition.* Participants were scanned using a 3 T Siemens MAGNE-TOM Skyra scanner (Siemens, München, Germany) at the Center of Brain, Behavior, and Metabolism (CBBM) at the University of Lübeck, Germany with 60 near-axial slices. An echo planar imaging (EPI) sequence was used for the acquisition of on average 1520 functional volumes (min = 1395, max = 1672) during each of the two sessions of the experiment, resulting in a total of on average 3040 functional volumes (TR = 0.992 s, TE = 28 ms, flip angle = 60°, voxel size = 3 × 3

× 3 mm³, simultaneous multi-slice factor 4). In addition, a high-resolution anatomical T1 image was acquired, which was used for normalization (voxel size = 1 × 1 × 1 mm³, 192 × 320 × 320 mm³ field of view, TR = 2.300 s, TE = 2.94 ms, TI = 900 ms; flip angle = 9°; GRAPPA factor 2; acquisition time 6.55 min; see Supplementary Fig. 10 for whole brain mask).

*FMRI data analyses.* FMRI data were analyzed using SPM12 (www.fil.ion.ucl.ac.uk/spm). Field maps were reconstructed to obtain voxel displacement maps (VDMs). EPIs were corrected for timing differences of the slice acquisition, motion-corrected and unwarped using the corresponding VDMs to correct for geometric distortions and normalized using the forward deformation fields as obtained from the unified segmentation of the anatomical T1 image. The normalized volumes were resliced with a voxel size of 2 × 2 × 2 mm³ and smoothed with an 8 mm full-width-at-half-maximum isotropic Gaussian kernel. To remove low-frequency drifts, functional images were high-pass filtered at 1/384.

Statistical analyses were performed using a two-level, mixed-effects procedure. A main GLM was implemented on the first level and this fixed-effects GLM included four epoch regressors modeling the hemodynamic responses to the different cue conditions (Ability: High vs. Low × Agent: Self vs. Other), weighted with the performance expectation ratings per trial as parametric modulator for each condition. Two regressors modeled the feedback conditions for Self vs Other collapsing across PE valence (Agent: Self vs. Other). Two parametric modulators were included per feedback condition, weighting feedback trials with PE valence (continuous effect of the signed PE values) and PE surprise (continuous effect of the unsigned PE values). Parametric modulators were not orthogonalized, thus each only explaining their specific variance. One regressor modeled the performance expectation rating phase. The estimation periods for Self and Other were modeled as two regressors, and emotion rating phases as separate regressor. Each of the regressors was modeled with the exact duration as presented during the experiment: The cue phase was modeled with a duration of 2.5 s, the expectation rating phase according to individual reaction times with a mean of 4.26 s (SD = 1.04), the estimation phase with 10 s, the feedback phase with 3 s, and the emotion rating phase with 22.51 s (SD = 3.85). To account for noise due to head movement, six additional regressors modeling head movement parameters were introduced and a constant term was included for each of the two sessions.

On the second level, beta images for the parametric weights of feedback were extracted for Self and Other. Four separate one sample t-tests were implemented for PE valence and PE surprise for Self and Other. For direct comparisons of PE valence and PE surprise responses for Self and Other, two repeated measures ANOVAs were conducted including the respective beta images for Self and Other. Differential tracking of the PE valence, depending on biased learning and self-conscious affect, were examined by three additional second-level models for the PE valence beta images for Self including either the Valence Learning Bias, embarrassment, or pride ratings as between-subject covariate. A self-related Pupil Dilation Bias (average slope for positive PEs - average slope for negative PEs; higher scores indicate stronger pupil dilation for positive PEs) was also included as covariate in another second-level model to assess whether the neural response scaling with more negative PEs was associated with the pupil dilation response. Here, we tested for stronger responses with more negative PEs associated with more negative affect and a more negative Valence Learning Bias and Pupil Dilation Bias. The analyses including all covariates were conducted within our predefined ROIs, the bilateral dAI, vAI, amygdala, mPFC, and VTA/ SN, as these regions are associated with affective and motivational aspects and PE tracking during learning (for a detailed description see section "Thresholding procedure and regions of interest"). Supplementary analyses assessing all four feedback conditions and parametric modulators are show in Supplementary Note 4, Supplementary Figs. 11–13.

We additionally performed psychophysiological interaction (PPI) analyses on the first level, investigating whether functional connectivity of the dAI, which is commonly activated during feedback processing independent of Agent and Prediction Error Sign (conjunction of baseline contrasts: feedback Self ∩ feedback Other), would differ depending on the PE valence. PPI analyses were computed separately for Self and Other and the resulting contrast images for the PPI effects were aggregated on the second level using two-sample t-tests contrasting PPI effects for Self vs. Other. For each participant, we defined 6-mm radius spherical ROIs, centered at the nearest local maximum for the conjunction contrast feedback Self ∩ feedback Other and located within 10 mm of the group maximum within the dAI, separately for the left dAI (x, y, z: −33, 20, −4) and right dAI (x, y, z: 36, 20, −7). By computing the first eigenvariate for all voxels within these ROIs that showed a positive effect for the conjunction (*p* < .500), we extracted the time course of activations and constructed PPI terms using the contrast for the parametric weights of PE valence for Self or Other, respectively, resulting in four distinct PPI first-level GLMs. One participant was excluded from the PPI analyses for the right dAI, because no voxels survived the predefined threshold for eigenvariate extraction. The PPI term, along with the activation time course from the (left or right) dAI was included in a new GLM for each participant that also included all the regressors in the initial first-level GLM (four regressors for the different cue conditions, each weighted with the expected performance ratings; two feedback regressors for Self and Other with each two parametric modulators for PE valence and PE surprise; two regressors for the estimation periods for Self and Other; one regressor for the expectation ratings phase; one regressor for the emotion ratings

phase; six regressors modeling head movement parameters; a constant term for each session). On the second level, we assessed whether there was a stronger functional coupling of the dAI seed regions with the predefined ROIs (amygdala, mPFC, VTA/ SN) for the Self in contrast to the Other when PE valence was more negative. In line with the negative Valence Learning Bias for self-efficacy beliefs and stronger pupil dilation responses scaling with more negative PEs we specifically tested for stronger functional connectivity correlated with more negative PEs. Functional connectivity dynamics were also associated with learning behavior by calculating Spearman correlations for the Valence Learning Bias and the mean parameter estimates for the PPI effect of Self > Other extracted from the GLMs described above in a sphere of 6 mm around the peak voxels within the predefined ROIs (amygdala, mPFC, VTA/ SN).

*Thresholding procedure and regions of interest.* According to its suggested role as an integrative hub for motivated cognition and emotional behavior, the AI was defined as one of the regions of interest (ROIs)[42,43]. Due to their specific functional associations, a bilateral ventral and a bilateral dorsal AI ROI was defined according to the three-cluster solution of Kelly and colleagues[47]. The bilateral amygdala was defined as another ROI and derived from the AAL atlas definition in the WFU PickAtlas[95] due to its similar role for the attention-emotion interaction[42,64]. The mPFC ROI was also derived from the AAL atlas in the WFU PickAtlas (label: bilateral frontal superior medial) due to its specific role during social learning and for biases in self-related learning reported in previous studies[33,96]. Additionally, an anatomically defined VTA/ SN ROI, dopaminergic nuclei in the midbrain, was included (probabilistic atlases of the midbrain; Adcock Lab)[97,98] as dopamine signals motivationally important events, e.g. during reward learning[45], and has been associated with biases in memory towards events that are of motivational relevance[44].

FMRI results were family-wise-error (FWE) corrected at peak level for the whole brain unless ROI analyses were conducted, and all coordinates are reported in MNI space. As our predefined ROIs were chosen with respect to their involvement with the emotion-cognition link, we tested the effects of our covariates on PE valence tracking and PPI effects within the ROIs. Anatomical labels of all resulting clusters were derived from the Automated Labeling Atlas Version 3.0[99].

**Reporting summary**. Further information on research design is available in the Nature Research Reporting Summary linked to this article.

## Data availability
The data that support the findings of this study are available from the corresponding author upon reasonable request. Source data for the figures are available as Supplementary Data 3.

## Code availability
Code used to generate the analyses are available from the corresponding author upon reasonable request. Software packages used for the analyses are RStan package (Stan Development Team, 2016. RStan: the R interface to Stan. R package version 2.14.1.), SPM12 (www.fil.ion.ucl.ac.uk/spm), GAMLj (https://github.com/gamlj/gamljin) in *jamovi* (Version 1.2.27, The jamovi project (2020), retrieved from https://www.jamovi.org).

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

## Acknowledgements

We would like to thank Prof. Christoph W Korn for his very helpful comments and discussions on the manuscript. We are also grateful to Clara Gunzelmann and Rebecca Rocksien for their help with data collection. The research was funded by the German Research Foundation (Temporary Positions for Principal Investigators: MU 4373/1-1; Sachbeihilfe KR 3803/11-1) and the Medical Department of the University of Lübeck (J21-2018). We would also like to thank the handling editors Stefano Palminteri and George Inglis as well as all three anonymous reviewers for their time and very helpful remarks.

## Author contributions

L.M.P., N.C., F.M.P. and S.K. designed the research. L.M.P. and N.C. acquired the data. L.M.P. analyzed the data and prepared the manuscript. L.M.P., N.C., A.V.M., A.S., D.S., F.M.P. and S.K. discussed the data analyses and interpretation of the results and reviewed and edited the manuscript.

## Funding

## Competing interests

The authors declare no competing interests.
