## [Peer Review File · Communications Biology]

Reviewers' comments:

Reviewer #1 (Remarks to the Author):

Müller-Pinzler et al. report a highly interesting and methodologically particularly elaborated study focusing on the omnipresent question of how affect/emotions interact with belief formation. The insights bring the field significantly forward and will be important for a broad audience. While I strongly recommend the publication, I do think that more work should be invested to increase the clarity of the manuscript, as described in detail below.

****Task design****:

The overall task idea and many details are exceptionally smart, elaborate and precise. However, several small technical details raised some questions. Did the authors conduct any kind of design efficiency tests to optimize the task design for fMRI? I was surprised by the long duration of the estimation phase of 10 sec. Was this really necessary relative to mean reaction time? And why was the performance expectation self-paced, but not the estimation? In addition, the jittered inter-stimulus-intervals after each trial event consumed a lot of time (please report the means for each of these, not only the range). Please report the mean reaction time for the estimation and make a statement for future studies whether the presentation time should be adjusted. Shortening the task would be beneficial for subjects' compliance and vigilance, and efficiency tests might help to optimally balance the fMRI design requirements and psychological compliance aspects.

****Data analyses****:

Overall, the data analysis follows high standards, as it combines model-free analyses, computational modeling including BMC, fMRI analyses including PPI, and the consideration of relevant covariates. However, some details could be added to enable a better understanding of the applied procedures, as listed below. In general, the clarity of the manuscript would benefit if analyses were reduced to the most necessary ones and if the authors would announce which analyses refer to the main hypotheses, which are needed to proof that these results of interest hold even when controlling for possible confounds, and which to provide some supplemental background information (e.g., average brain responses to feedback Self > Other).

* Behavioral data analysis: I was wondering why on some occasions repeated measures ANOVAS (e.g., where trials could also have been modeled as a random effect in a linear mixed effect model, see lines 524-526), and on other occasions mixed effect models were used (e.g., lines 627-635). Without explaining the choice of different tests this appears somewhat arbitrary. Lines 627-631: Please specify the random structure more precisely: did you model a random intercept or a random slope? And did you fit one model per subject in order to obtain the subject-wise association between PE and pupil slope, or did you include all subjects into one model? Lines 631-635: This is also underspecified: What was the dependent variable in this model, the betas from the subject-level LME? In general, providing the model formula would provide more clarity.

* Pupil data analysis: The authors report that only the Self-condition was analyzed because "onsets during feedback strongly differed between Agent conditions". I don't understand this explanation. What onsets are here referred to? The duration of the feedback phase was always 3 sec and the pupil slope within this time interval should be generally comparable between self and other.

* First-level fMRI GLMs:

* i) The authors explain that they modeled the following trial events on separate regressors (including further condition-specific differentiations): CUE, ESTIMATION, FEEDBACK, EMOTION RATING, INSTRUCTION. Please specify more precisely: What were the respective durations of these events - 0 or the duration of the presentation on the screen? Was the performance expectation rating event modeled on a separate regressor?

* ii) Was the aim of the first GLM to identify average BOLD responses to feedback, dependent on

valence and agent? Why was then parametric modulation applied to CUE and FEEDBACK if not needed to achieve this aim?

* iii) The authors specify 2 parameters for the parametric modulation of feedback: PE and absolute PE. Please specify whether these parameters were orthogonalized or not and what consequences this has for existing shared variance. It may even be a good idea to simply specify separate 1st-level models, the one with parametric modulation of feedback by PE, and the other with parametric modulation of feedback by absolute PE, in order to avoid uncertain handling of shared variance. Alternatively, an orthogonalization with an alternating order of parameters (in two separate GLM, respectively) would allow for the assignment of the shared variance to the first parameter with the opportunity to report the unique variance of the second parameter (see Mumford, 2015).

* iv) I'm not convinced by the additional value of the third model. PE is a difference between feedback and performance expectation, and it does not help to show the separate effects of these components because they represent different psychological constructs. If you want to control for the feedback level (or performance expectation), then better include the feedback level as a first parameter and PE as a second, orthogonalized parameter within a parametric modulation of feedback, and report the effect of PE (see Mumford, 2015).

* Second-level GLMs: The whole paragraph (lines 677-689) is not easy to follow. It is not clear i) how many different models were specified, ii) what contrast images each of them included (PE/absolute PE or others), iii) why sometimes flexible factorial and sometimes t-test were used, iv) was only one contrast image for Self included in the "additional second-level models" (line 683) and only one covariate per model? v) Relating to the covariate-GLMs: Please specify that (according to Table S7) and why these analyses were restricted to ROIs.

* PPI analysis:

* i) What does it mean to "assess whether there was a stronger functional coupling of the dAI with our predefined ROIs ... when PE valence [pos arrow neg] was more negative"? Did the authors test this via the contrast -1 to identify connectivity that would be stronger the smaller PE was? Please explain why you focused on the negative effect specifically. Was the opposite contrast tested as well (stronger connectivity during larger, i.e. positive PE)?

* ii) Lines 715-718: Was this a separate GLM with Valence Learning Bias included as a covariate? Which predefined ROIs do you refer to here: dAI and/or others?

* ROIs: Please specify whether VTA/SN was an anatomically specified ROI and if yes based on which atlas.

* fMRI significance threshold:

* i) "FWE corrected on the whole brain level" (line 732-733, plus e.g., Table S5) is underspecified - you can apply the FWE correction at the cluster or peak level, and both can relate to the whole brain or to a small volume. Please specify more precisely.

* ii) Please use the same statistical threshold for all brain analyses (e.g., FWE-corrected at the peak level + extent threshold, or FWE-corrected at the cluster level + cluster-defining threshold), or clearly justify why the threshold needs to be adjusted for the different analyses.

* iii) Before applying the small volume correction for ROI analysis, which whole brain significance threshold was applied?

****Results:****

* BMS: Please report the expected frequency in addition to p_{xp} in order to indicate the effect size.

* Behavioral analyses: Please report effect sizes as well.

* fMRI results are not communicated clearly enough with respect to their technical description and their interpretation, please double check and correct through out the manuscript. Arbitrarily selected examples just to make the point clear (not a complete list):

* i) Supplementary, lines 49-50: "Parameter estimates for the differences between feedback for positive vs. negative PEs derived from the peak voxels of the interaction effect depict the interaction in the left NAcc/VS ...". Why not something like: "Parameter estimates correspond to the difference in BOLD response to positive relative to negative PEs in bilateral NAc [MNI coordinates]. The plot shows that positive relative to negative PEs increased the activity in NAc only when learning about the own

performance, but not when observing others." In addition, the plot title rather refers to the y-axis than to the general plot, that additionally differentiates between self and other. Furthermore, plotting 4 different contrast estimates (self-PE-pos, self-PE-neg, other-PE-pos, other-PE-neg) would provide a more complete information about the interaction effect (e.g., was there a deactivation in response to self-related negative PE and an activation in response to self-related positive PE?).

* ii) Table S4: "... associated with the parametric weights of ..." is a verbal double dipping. "Activation associated with (or correlated with) absolute PEs" would be enough. The anatomical label "Temporal Pole/*" seems to be displaced and to refer to the cluster below (size 107). And what do you mean with "Cluster extends refer to ..."? Do you mean "cluster extents"? Still, this is an unusual wording to specify the significance level, and makes only sense for FWE-correction at the cluster level (e.g. same wording is used in Table S5, with the additional note that "Only clusters with more than 50 voxels are reported...").

* iii) Table S7: "Covariates Associated with Individual Differences ..." again verbal double dipping. "Covariates revealing individual differences..." for instance sounds better.

* iv) I would suggest not to use the quite unusual label "PE valence [neg arrow pos]" to describe the parametric modulation of the BOLD response to feedback by PE. This is confusing, as (here or in other analyses) the authors simply refer to (signed) prediction errors, which is a parameter well established and often reported in the literature. To make things worse, the authors also use "PE Valence" (Table S5) and "Valence Specific Prediction Error Tracking" (Table S7) to refer to the same effect, and "PE valence [pos | neg]" to refer to a different, categorical variable. Wouldn't it be easier to simply refer to PE, absolute PE ("PE surprise"), and PE valence (categorical variable, as valence has only two levels, positive and negative) to differentiate these effects?

* v) Be careful when saying that the activity increased in response to negative vs. positive PEs (e.g., line 290) when referring to the parametric modulation of the BOLD response to feedback by PE. "Negative vs. positive" sounds like a categorical variable (disregarding the size of the PE, and taking into account only its valence), but you actually refer to a correlation between 2 parameters. Therefore, you need to say that the activity scaled with increasing (decreasing) PE, or positively (negatively) correlated with PE, etc.

****Discussion**:**

* Lines 403-404: Why is the "common currency" assumption related to absolute PE ("PE surprise")? Cited Ruff and Fehr (2014) and other relevant publications specifically refer to value computation: "neural value signals that encode the rewarding properties of the choice options". This is related to the signed PE, where large positive numbers reflect positive values and small negative numbers reflect negative values. Instead, large absolute PE reflect both positive and negative values and therefore cannot be validly related to theories and findings about value computation.

* Lines 409-411: What exactly do you refer to: The modulatory effect of absolute PE and the one of PE? This comparison is not valid as the two parameters relate to different psychological constructs, and not merely to different extents of balance.

* Could you shortly comment on why you think people rely more strongly on negative PEs when learning about their efficacy, but do the opposite when for example estimating future risks as reported in the "optimism bias" literature?

****Minor comments**:**

* Was it on purpose that the word "affected" in the title is open to interpretation? It could mean both affected in the sense of influenced by something, and affected in the sense of affective or emotional. If not, "affective" instead of "affected" might be more straight forward. While the term is explicitly introduced in the manuscript (line 73, however not in the abstract), the title may be difficult to understand without this explanation.

* line 64-66: another relevant reference could be Kunda, 1990, The Case for Motivated Reasoning (optional)

* line 69 and others: another relevant reference could be Vinckier et al., 2018, Neuro-computational

account of how mood fluctuations arise and affect decision making

* What do you mean with "average number" (e.g., line 492)?

* Figure 1: The legend says that the cue indicated both the estimation category and the agent, but the figure and the methods indicate that it was only the category. Furthermore, why was the format of performance percentile scale changed from the expectation rating (white line) to the feedback (colored bar) instead of using one format for both scales? The method text says that it was the same scale, but the screen shots in the figure indicate a changed format.

* Please report whether the group mask in 2nd level fMRI analyses covered the whole brain or whether some areas (e.g., vmPFC and ventral striatum) were not completely included due to susceptibility artifacts, which may be more pronounced in multiband MR sequences.

* Please use inconsistent abbreviations across the manuscript, e.g., correct inconsistent condition abbreviations in lines 606-607 vs 612-613.

* Maybe I missed something, but just to be sure (line 361-362): Have you reported the final efficiency beliefs as well (or the total change from the initial to the final belief, across the entire experiment)?

* Lines 386-388: What do you mean with idiosyncratic response patterns?

* When discussing the functional connectivity of the dAI, can you also provide some evidence on underlying anatomical connectivities that would support the assumed interpretation?

Reviewer #2 (Remarks to the Author):

In this manuscript, Müller-Pinzler and colleagues use a belief updating task in conjunction with self-report ratings of affect, computational modelling, pupil dilation and fMRI to characterise information integration in response to differentially valenced (negative, positive) information. Behaviourally they show that beliefs about oneself are updated asymmetrically with greater updating in response to negative vs positive information. This is nicely unpacked using computational modelling by fitting separate learning rates for positive ($\alpha+$) and negative ($\alpha-$), revealing a difference in the size of these rates ($\alpha- > \alpha+$). This learning bias is shown to correlate with subjective self-reports of positive (pride) and negative (embarrassment) with this bias (towards negative over positive learning) being enhanced with higher embarrassment and reduced for higher pride ratings. Pupil dilation during information presentation was higher for negative compared to positive PEs (controlling for unsigned PEs, i.e. general surprise/saliency). fMRI suggested that the mPFC responded to the general salience of the information (unsigned PEs) independent of valence and the self/other frame. Interestingly, responses in the NAcc, a region well known to respond to rewards such as money, were much more selective - responding to signed PEs specifically for self relevant information, consistent with the idea that information itself can be rewarding and is more rewarding the better than expected it is. The authors also examine how the parametric effect of signed PEs interact with a number of individual difference measures (pupil dilation, learning rates, subjective ratings) in a number of ROIs (amygdala, insula, MPFC and VTA/SN).

I am convinced by the behavioural results and computational modelling which replicates previous work from this group using this task (2019). However, I was a bit less convinced by the some of the results that followed this which made great efforts to try and link this interesting behavioural finding (asymmetry in updating) to self report, pupil dilation and fMRI measures. I detail these concerns below.

Ratings

Why did the authors choose to get participants to rate "embarrassment" and "pride" specifically? It seems the goal was to tap into a subjective sense of participants positive and negative affect, so why narrow it down to asking about these emotions specifically? Would it not have been easier to ask how happy/positive (vs unhappy/negative) the participants was? In other words, how much do the authors think participants are submitting ratings about these feelings specifically; would they expect the same results if they rated joyous/disappointed/satisfied/unsatisfied etc.

I was not clear what emotions participants rated for each condition. Very simply, could the authors state for each of the 4 conditions (high ability self, low ability self, high ability other, low ability other) which emotions (Embarrassment, Pride, Happiness, Stress/Arousal) participants were asked to rate, how many times they rated each of these emotions and when they rated them (e.g., start/middle/end of a condition). If specific emotions (e.g., Pride) were probed for specific conditions (e.g., high ability self) is there a risk that this could cue participants about the conditions?

If participants only rate each emotion once (or twice), could it be that the emotions being rated are not anything to do with the task and the information participants are getting during it but is much more general (i.e. how they generally feel).

Pupil dilation

I am not an expert in pupil dilation analysis. But with that caveat, some potential issues that occurred to me are:

(1) The task is not optimised to look at pupil dilation. Did the authors control for things like luminance, distance from the monitor, etc.

(2) The authors regress mean pupil dilation during feedback on each trial against signed and unsigned PEs (both in the same model) in a mixed effects model. Even though the two regressors will be completely uncorrelated (as half will be perfectly positively correlated and the other perfectly negatively uncorrelated rendering the overall correlation 0), this seems like it will create an odd tiling of the space whereby you have no instances in which unsigned PEs ("surprise") are high and signed PEs ("valence") are low and also no instances where unsigned PEs ("surprise") are low and signed PEs ("valence") are high. Is that right? Have the authors done any simulations to guard against the possibility that this might create some odd effects.

(3) Why did the authors only include the signed PEs (and intercepts) as random (per subject) effects but not the absolute PEs (which are only in there as fixed effects)? I could not see any rationale for this choice of model but maybe I missed it.

(4) A "classic" pupil dilation response peaks around 2 seconds after onset (e.g., of information). In the past I've seen researchers take this peak (or peak minus dilation at trial onset) as their trial by trial measure of dilation (rather than taking the average the entire response over the trial. But the pupil response in the authors data looks rather different to this (orange trace in Figure 2b) – why is that the case? And what is the rationale for averaging responses over the entire window (e.g., versus taking the peak, peak minus trough, etc.)?

(5) [related to point above] The shape of the average pupil diameter trace (Fig 2b) made interpreting the results a bit tricky for me. The authors report that "more negative PEs are associated with greater pupil slopes compared to positive PEs" (Figure 2b). That's true, but the authors interpret this as being consistent with the negative learning bias they find in the computational model. But since all slopes (for positive and negative PEs) are negative in their analysis, another way of saying this is that negative PEs are associated with flatter slopes compared to positive PEs (which have much steeper slopes). And one could conceivably think of ways that steeper slopes might indicate greater arousal (e.g., dropping back down to baseline from a higher peak). This is not to say that the authors interpretation is necessarily wrong, it just seems to me that you could interpret the direction of this result in various ways...

fMRI

Line 259-261: I do not agree that the absence of a significant difference between self and other suggests that there is a common process of error tracking independent of the agent (lack of an effect for a difference between the conditions is not evidence that there is no difference).

I found a lot of the results here a bit difficult to join up with the behavioural/computational results that had gone before it. I think this is in part because the fMRI model uses a conceptually different approach to the computational modelling. Specifically, the learning model effectively separates trials into positive and negative information trials and uses one of these two learning rates in the update rule (depending which of these two trial types applies). The fMRI model does not do this. It collapses all trials together and uses a signed PE to identify which areas of the brain BOLD scales with. But it could be that different regions respond to positive and negative information (e.g., region X correlates with the magnitude of positive PEs and region Y the magnitude of negative PEs) which I don't think would be detected using this approach.

[sidenote: this approach also makes interpreting the later analysis - with predefined ROIs - tricky and I am not sure I agree with some of the interpretation as it is. E.g., the authors interpret (lines 288-292) that the more negative the learning bias is, the more neural activity responds to negative vs positive (see Figure 4d, left panel). But I don't think that is quite right. If I understood the analysis correctly, it would be more accurate to say that the more negative the learning bias is, the less sensitive neural activity is (i.e. the weaker the parametric effect is) in these regions to the valence of the PE (but it may be for instance that at all levels of learning bias, the sign of the parametric regressor is positive, i.e. greater activity for positive PEs relative to negative PEs). The same interpretational issue applies to other analysis using individual difference measures (ratings, pupillometry).]

To address some of this, I think it would be good if the authors modelled the fMRI data with a different GLM in which, instead of having signed PEs and absolute PEs as parametric regressors (pmods) in the same GLM (modulating the timepoint information is presented), they instead have only absolute PEs as pmods but split the time information is presented into two events "positive valence" events and "negative information" events. That way, it might be easier to identify (a) the different and overlapping brain regions for each type of valence, (b) contrast parametric effects of positive with negative, (c) relate activity to learning rates and subjective ratings.

Reviewer #3 (Remarks to the Author):

Review of Neurocomputational mechanisms of affected beliefs by Müller-Pinzler et al.

Summary

In this study, the authors investigated the neurocomputational mechanisms underlying the interplay between beliefs formation about self-efficacy and affects. To that end, they analyzed the behavior, pupil data and BOLD responses of 39 subjects (and additional behavior of 30 other subjects) performing a task involving a series of cognitive estimations (e.g., estimating the weight of animals) and efficacy predictions. Feedback on efficacy was manipulated such that a low ability condition (negative feedback) and a high ability condition (positive feedback) were created. Subjects could perform the task themselves in one condition or predict the efficacy and observe the behavior of others in a second condition. Different learning models were fitted to the data to highlight learning mechanisms. Authors found that subjects tended to take more into account negative feedback about their self-efficacy when they performed the task themselves whereas they learnt in an unbiased manner when they observed others perform the task. They also found that this learning bias was related to experiences of pride and embarrassment, pupil size and neural activity.

The question investigated in the present study (i.e., what are the neurocomputational bases of "affected beliefs" formation?) is of prime interest, and numerous and various analyses have been used to answer that question. Overall, the analyses are clear and well written, and figures are precise. However, behavioral, neural and pupillometry hypotheses are not very well stated in the current

version of the manuscript and the paper tends to look like a collection of analyses without clear justifications. I also have a series of major and minor comments, particularly on the behavioral and computational parts.

Major Comments

#1 LOOP task and predefined prediction errors

#1.1 In the task used in this study, feedback is manipulated such that a series of predefined prediction errors is presented to subjects, instead of a series of predefined outcomes (as in previous versions of the task). This feature implies that the more extreme subjects' expectations are, the even more extreme the feedback will be, and in a similar fashion, the more subjects update their expectations (high learning rate), the more extreme the feedback becomes. It would then be interesting to see how actual feedback differed between subjects. Was the feedback received by biased subjects more extreme (close to 0) in the low ability condition versus the high ability condition (close to 100)? Was the negative feedback closer to 0 for biased subjects versus unbiased subjects?

#1.2 It is of prime importance insofar as pride and embarrassment ratings could potentially be linked to the more extreme feedback received rather than to the higher learning rate for negative PE. Could the authors comment on this?

#2 w parameter in the winning model

#2.1 In the winning model (M8), learning rates are reduced for the most extreme feedback. It is not clear how this parameter interacts with learning rates themselves. Simulations of the winning model could help understand how learning rates are impacted by different values of w . Also, it would be useful to see the fitted w parameters for each group of subjects (low vs. high bias) or potentially the learning rates from the Valence model without w (Model 5). Is w higher in the high bias group?

#2.2 As suggested in major comment #1, if more biased subjects obtained more extremely negative feedback, which are among those the most affected by w , it is not clear whether the effect of a higher learning rate for negative feedback is compensated or not by the effect of w .

#3 Expectations initialization

The learning curves in figure 1a look extremely similar (self vs. other) with differences mainly in the initial predictions. In the task, initial expectations are set to 50% to compute manipulated feedback while in the models, they are set as free parameters. Could the authors explain the rationale behind this parametrization? Value initializations seem important when evaluating learning rate asymmetries and it would be interesting to check whether this difference in value initialization could explain in part the difference in learning rates.

#4 Affect measures

#4.1 Affect scores used to categorize subjects are computed as the average of only two measures per condition (self vs. other). Furthermore, the only difference between those measures is the condition subjects were in in the previous trial. Could authors explain why they think that this score represents uniquely affects for one of the two conditions (self or other)?

#4.2 In the manuscript, it is repeatedly stated that emotional states shift preferences for information of positive or negative valence (e.g., lines 117-118). However, measures of emotional states are recorded after learning phases. It would be interesting to fit the model separately before and after these emotional states measures and look at the difference in learning rates at these different times. Could authors comment on this?

Minor Comments

#1 Figure 2d and 2e would benefit from a harmonization of the Y-axes in the scatter plots and line plots respectively. The plots are difficult to compare because of their different Y-axis.

#2 It is not clear why 30 additional subjects have been added to the study and the difference in sample size between the behavioral / computational analyses and the pupillometry / neural analyses is not very well stated.

#3 There's a repetition line 235/236:

234 It has been suggested that pupil dilation reflects differences not only between stimuli
235 but similarly between individual biases during decision making (see Figure 2d for examples
236 of individual differences (see Figure 2d for examples of individual differences)

Reviewers' comments:

Reviewer #1 (Remarks to the Author):

Müller-Pinzler et al. report a highly interesting and methodologically particularly elaborated study focusing on the omnipresent question of how affect/emotions interact with belief formation. The insights bring the field significantly forward and will be important for a broad audience. While I strongly recommend the publication, I do think that more work should be invested to increase the clarity of the manuscript, as described in detail below.

****Task design**:**

Comment #1a: The overall task idea and many details are exceptionally smart, elaborate and precise. However, several small technical details raised some questions. Did the authors conduct any kind of design efficiency tests to optimize the task design for fMRI? I was surprised by the long duration of the estimation phase of 10 sec. Was this really necessary relative to mean reaction time?

- We very much agree with the reviewer that the task setup is not the most efficient version in terms of fMRI design. The critical argument to implement the long estimation phase was to convey to the subjects as credibly as possible that the feedback relates to their efforts to solve the task. To avoid random guessing, subjects needed to have sufficient time to develop and (re)evaluate their estimate and move the pointer to the desired position. Only when this is true can subjects relate the feedback to their own performance and learn from it with respect to their self-efficacy beliefs. To achieve this, we have accepted that the efficiency of the design is not quite optimal with respect to the fMRI but knew from prior applications of a comparable design, that the outcome period can be well modeled after longer estimation periods resulting in plausible signal (REF embarrassment). Response times in the current study confirm that participants indeed used the long duration of 10 sec to solve the task. When excluding all missings (which should at least partially be due to too

slow response behavior) mean reaction times were 7.15 sec (SD: 0.85 sec) for the first click/ response selection. Participants were allowed to correct their response after the first click until the 10 sec were over, unfortunately, we did not track this reaction time. This suggests that participants on average took at least 7.15 seconds or longer to answer the questions and from our post-experimental interviews and experience with the task, we suspect that participants might have even taken longer than 10 sec if responses were self-paced. We also tested a different version of the LOOP task in another (non-fMRI) study during which participants had to answer all questions beforehand and only received feedback on their previous answers during the actual learning task. However, in our experience, without having the opportunity to put more effort into the next question directly after receiving feedback, participants were less emotionally involved during the task and were less likely to attribute the feedback to their abilities in estimating the different properties, which was critical for the current study.

Comment #1b: And why was the performance expectation self-paced, but not the estimation?

- As mentioned above we think that self-paced responses might be even longer than 10 seconds and prolong the whole task even more in case of the estimation period. The current design was thus a compromise to balance the trade-off between design efficiency and validity of our data and cover story. Extreme interindividual differences in the durations of the estimation phase could have made fMRI analyses more difficult if durations were to be correlated with previous feedback or feedback conditions (e.g., participants try to use more time after negative (or positive) feedback for solving the task). As the assessment of the performance expectations was unrelated to the effort execution during estimation we did not (i) observe any systematic variability that could compromise the validity of our fMRI design and (ii) unreasonable extension of this period in a self-paced design. In addition, in contrast to the estimation period, where we could induce a little time pressure without invalidating the cover story, our experimental design hinges on an accurate

assessment of the subjective performance expectations. This reason finally made us decide to use a self-paced assessment as it was deemed necessary in order to avoid confounds due to e.g., greater uncertainty about the own performance (greater uncertainty should not result in additional (un)systematic error in the assessment due to greater time pressure). Still, even for the self-paced rating of the performance expectation subjects were much faster compared to the cognitive estimation 4.26 sec (SD=1.04) which was in line with our prior experience.

Comment #1c: In addition, the jittered inter-stimulus-intervals after each trial event consumed a lot of time (please report the means for each of these, not only the range).

- We thank the reviewer for this question. Inter-stimulus-intervals did consume a lot of time, but these were necessary so that each of the following task phases could be analyzed without the BOLD response of the previous phase systematically leaking into the following phase of interest. We agree that we did not describe the inter-stimulus-intervals in detail and thus added the following description:

Lines 524-27: "Jittered inter-stimulus-intervals were presented following the cue (mean: 4 * TR (0.992 secs), range: 2-6 * TR), estimation (mean: 4.5 * TR, range: 2.5 – 6.5 * TR) and feedback phase (mean: 6 * TR, range: 4-8 * TR) for the fMRI task with jitters distributed in a uniform distribution with steps of 0.5 * TR."

Comment #2: Please report the mean reaction time for the estimation and make a statement for future studies whether the presentation time should be adjusted. Shortening the task would be beneficial for subjects' compliance and vigilance, and efficiency tests might help to optimally balance the fMRI design requirements and psychological compliance aspects.

- Thank you for this remark and considerations for future design optimizations. Shortening the task will be very important specifically when sampling from significantly younger or less healthy populations. Mean reaction times for the estimation were 7.15 sec (SD: 0.85 sec) for the initial response selection, and as mentioned above we, unfortunately, did not record response times for

adjustments following the initial response selection. Therefore, as discussed above, we don't think that the presentation of the estimation phase could be shortened extensively. But we agree with the reviewer that we could look for other options to shorten the task. One option for future versions of the task could be to potentially shorten inter-stimulus-intervals before the Cue phase, as we did not focus on analyzing the Cue phase. We could also run the whole task without the Other condition in future studies as we were able to assess now, how self-related learning differed from other-related learning in terms of learning biases and neural correlates. For the purpose of our primary fMRI study to implement the LOOP task, we aimed to get a more or less complete picture and therefore decided to run a rather long version. Also, the other-related learning part was important to have the opportunity to draw conclusions with respect to the specificity of the self-related biases. But we totally agree that shortening the task for future studies will be beneficial for compliance and vigilance.

****Data analyses**:**

Comment #3: Overall, the data analysis follows high standards, as it combines model-free analyses, computational modeling including BMC, fMRI analyses including PPI, and the consideration of relevant covariates. However, some details could be added to enable a better understanding of the applied procedures, as listed below.

In general, the clarity of the manuscript would benefit if analyses were reduced to the most necessary ones and if the authors would announce which analyses refer to the main hypotheses, which are needed to proof that these results of interest hold even when controlling for possible confounds, and which to provide some supplemental background information (e.g., average brain responses to feedback Self > Other).

- We agree with the reviewer that analyses should have been structured in a better way. In line with the suggestion to reduce the analyses in the main manuscript to the most necessary ones, we moved several analyses to the Supplement (e.g. average brain responses to feedback Self > Other) and

removed some analyses entirely (e.g. the third GLM assessing neural activity in response to different components of PEs). We also specifically stated our hypotheses at the beginning of each section (behavior, pupil, brain) which has also been suggested by reviewer #3 who had similar concerns regarding the clarity of the presentation of our results.

* Behavioral data analysis:

Comment #4: I was wondering why on some occasions repeated measures ANOVAS (e.g., where trials could also have been modeled as a random effect in a linear mixed effect model, see lines 524-526), and on other occasions mixed effect models were used (e.g., lines 627-635). Without explaining the choice of different tests this appears somewhat arbitrary.

- We agree with the reviewer that the choice of different models seems a bit arbitrary even though the different implementations should result in similar outcomes. To streamline our analyses, we have now replaced the repeated measures ANOVAs for expectation ratings and predicted data (Supplement) with linear mixed models. These analyses are now more in line with the analyses of the pupil data. We adapted the corresponding sections in the Supplementary Methods and Results section, where both analyses have been moved to.

Supplement Methods Lines 87-92:" To illustrate effects in our behavioral data, a model-free analysis was performed on the participants' expected performance ratings for each trial. We conducted a linear mixed model (LMM) fitted with a restricted maximum likelihood (REML) including the Ability condition (High Ability vs. Low Ability) x Agent condition (Self vs. Other) as factorial and Trial (20 Trials) as continuous predictors. Intercept, Ability condition, Agent condition, and Trial were modeled as fixed and random effects."

Supplement Results Lines 145-156: "The Trial x Ability condition x Agent condition linear mixed model revealed a significant main effect of Ability condition ($\beta=-20.54$, $t_{(68)}=-13.45$, $p<.001$, 95% CI=[-23.54; -17.55]) and interaction of Trial x Ability condition ($\beta=-1.23$, $t_{(5240)}=-36.55$, $p<.001$, 95% CI=[-1.30; -1.16]), indicating that participants adapted their performance expectation ratings over time according to the feedback provided in each Ability condition (see Figure 1c). Moreover, there was a significant main effect of Agent condition ($\beta=6.78$, $t_{(68)}=6.52$, $p<.001$, 95% CI=[4.74; 8.82]), indicating that participants evaluated their own performance more negatively than the other's performance. There was also a significant interaction of Agent condition x Ability condition ($\beta=1.29$, $t_{(5240)}=3.33$, $p<.001$, 95% CI=[0.53; 2.05]). The three-way interaction of Trial x Agent condition x Ability condition ($\beta=0.18$, $t_{(5240)}=2.66$, $p<.001$, 95% CI=[0.05; 0.31]) revealed a significant effect, hinting at differential learning patterns between the Ability conditions for Self vs. Other."

Comment #5: Lines 627-631: Please specify the random structure more precisely: did you model a random intercept or a random slope? And did you fit one model per subject in order to obtain the subject-wise association between PE and pupil slope, or did you include all subjects into one model? Lines 631-635: This is also underspecified: What was the dependent variable in this model, the betas from the subject-level LME? In general, providing the model formula would provide more clarity.

- We thank the reviewer for this comment. We agree that the description of statistical analyses might have missed some more details to avoid misunderstandings. The depicted formula only contained the fixed-effects for illustration purposes. To clarify, the analyses of the Pupil data were two-fold. First, we calculated the pupil slope of each trial for each subject to obtain a measure for the trial specific change in pupil diameter. The pupil slope was then passed as the dependent variable in the LMMs. Here, each LMM was fitted including intercept, PE valence, and PE surprise both as fixed and random

effects. The other three models were similar (i.e. pupil slopes as dependent variable) just adding another second level covariate. We added the description in the methods section, highlighted in the figure caption that the formula does not contain random-effects, and have added the full formula of the LMM in the method section. Due to another comment below we also modeled PE surprise and PE valence both as random effects.

Lines 652-657: "The linear mixed model (LMM) was fitted including intercept, PE valence, and PE surprise both as fixed and random effects. Additionally, this model was extended by including either embarrassment ratings, pride ratings, or the Valence Learning Bias as second-level covariates as well as their interaction with PE valence in three separate LMMs (see Figure 2c and Supplementary Methods for model description)."

* Pupil data analysis:

Comment #6: The authors report that only the Self-condition was analyzed because "onsets during feedback strongly differed between Agent conditions". I don't understand this explanation. What onsets are here referred to? The duration of the feedback phase was always 3 sec and the pupil slope within this time interval should be generally comparable between self and other.

We agree with the reviewer that this section could have been better described. Part of the problem was our misleading use of "onset" together with an insufficient explanation and visualization of the issue. The issue in the difference between the Self and the Other condition is that the offset on the y-axis (i.e. the pupil diameter) at the onset of the trace at the beginning of the feedback presentation is very large. This is reasonable and very much expected based on our understanding of the pupil dynamics in response to effort and arousal. The problem is, that this results in differently biased estimates of pupil slope, which is our summary measure for the change in pupil diameter during the course of the feedback presentation and confounds the comparison

between the conditions. We want to elaborate on our thoughts on that issue below.

The reason for the offset at the beginning of the feedback presentation is due to the greater arousal during the execution of the Self task compared to the Other condition. It is well known that cognitive demanding tasks and emotional arousal dilate the pupil, due to locus coeruleus (LC), noradrenergic innervation (Joshi et al. 2016). This arousal-related pupil dilation is sustained and the intensity and duration of the dilation are associated with the stimulus intensity (Geuter et al. 2014). Finding the pupil to be much wider at the beginning of the Self-feedback compared to the Other-feedback, is thus not surprising at all as participants were more emotionally engaged and executed a more demanding task prior to the feedback. The effect is shown in the figure below, with this offset being much larger than the effect of interest on positive or negative feedback.

As the pupil slope is a robust measure of relative change during stimulus presentation (i.e. greater decrease/increase not being sensitive to the shape or specific outliers, see also (Laura Müller-Pinzler et al. 2015; Einhäuser 2017; Geuter et al. 2014)), it, however, is difficult to interpret if the offsets are significantly different. First, since in all our analyses pupil diameter decreases during stimulus presentation, due to e.g. lower luminance of the feedback frame, slopes are negative on average. Greater arousal during feedback presentation should nonetheless result in less negative slopes, as we have shown for PE Valence and PE Surprise. When comparing the Self and the Other condition, however, it would seem that the arousal for the Other condition is higher, as slopes are less negative (see Figure R1). However, this would be misleading as the more negative slope during the Self condition is only due to the greater pupil at the beginning of the trial. Second, pupil constriction and dilation is limited by the physiology of the eye. This results in biases when comparing slopes that have largely different onsets since when the pupil diameter is larger at the beginning it is more limited in further dilation and less limited in further constriction compared to a trial starting with a smaller

diameter. From our point of view, these biases render the comparison between the Self- and the Other condition invalid as even the comparison of the PE-associated dynamics are confounded by the offsets at the trial starts.

Lines 645-652: “Pupil traces were only analyzed for the Self condition as there was a significant offset in the pupil diameter at the beginning of the feedback presentation. The strong difference in the pupil diameter between the Agent conditions is expected given the greater arousal (Joshi and Gold 2020) after the cognitive effort when estimating quantities. While the pupil slope is a robust measure of rather sustained relative change during stimulus presentation, this offset in the diameter at the beginning of the feedback presentation makes it impossible to draw valid comparisons of these slopes between the Agent conditions as greater pupil diameter will result in greater negative slopes compared to smaller pupil diameter.”

Figure R1. Pupil diameter traces for the feedback phase (3 secs) for all four feedback conditions uncorrected for offsets at feedback start. The bold lines show pupil traces for Self and the dashed lines for Other. Red lines show pupil traces for feedback with positive PEs and blue lines for negative PEs.

* First-level fMRI GLMs:

Comment #7a: * i) The authors explain that they modeled the following trial events on separate regressors (including further condition-specific differentiations): CUE, ESTIMATION, FEEDBACK, EMOTION RATING, INSTRUCTION. Please specify more precisely: What were the respective durations of these events - 0 or the duration of the presentation on the screen? Was the performance expectation rating event modeled on a separate regressor?

- We added a more detailed description of the durations we used to model each of the trial phases:

We also thank the reviewer for pointing out that we missed mentioning the performance expectation rating phase. This phase was modeled as a separate regressor and we included this in the revised version of the manuscript.

Lines 688-694: "One regressor modeled the performance expectation rating phase [...] Each of the regressors was modeled with the exact duration as presented during the experiment: The cue phase was modeled with a duration of 2.5 secs, the expectation rating phase according to individual reaction times with a mean of 4.26 sec (SD=1.04), the estimation phase with 10 secs, the feedback phase with 3 secs, and the emotion rating phase with 22.51 sec (SD=3.85)."

Comment #7b: * ii) Was the aim of the first GLM to identify average BOLD responses to feedback, dependent on valence and agent? Why was then parametric modulation applied to CUE and FEEDBACK if not needed to achieve this aim?

- As suggested by the reviewer, the aim of the first GLM was to identify average responses to feedback specific to the experimental conditions. The parametric weights are therefore not specifically necessary. We, however, aimed to reduce error variance at the first level by introducing them. Introducing parametric weights also does not impact the effects of the condition-specific contrasts as they are orthogonalized to the condition regressors.

Comment #7c: * iii) The authors specify 2 parameters for the parametric modulation of feedback: PE and absolute PE. Please specify whether these parameters were orthogonalized or not and what consequences this has for existing shared variance. It may even be a good idea to simply specify separate 1st-level models, the one with parametric modulation of feedback by PE, and the other with parametric modulation of feedback by absolute PE, in order to avoid uncertain handling of shared variance. Alternatively, an orthogonalization with an alternating order of parameters (in two separate GLM, respectively) would allow for the assignment of the shared variance to the first parameter with the opportunity to report the unique variance of the second parameter (see Mumford, 2015).

We thank the reviewer for this remark and apologize for not providing this information. PE and absolute PE were not additionally orthogonalized in the fMRI model. Notably, with this GLM specification, we model the unique, additive effects of the PEs independent of the effect of the other (i.e. signed PE while controlling for unsigned PE and unsigned PE while controlling for signed PE). We think this is important as our question of interest is whether the hemodynamic response scales with the PE in the sense of its valence (signed PE) or with the absolute deviation from the expectation in the sense of surprise (unsigned PE). Even though the regressors are rather independent already - due to our experimental design and due to the centering - this remaining shared variance would confound the interpretation of the results of the fMRI analyses. In fact, a change in BOLD signal could be wrongfully assigned to either scale with signed or unsigned PE when not controlling for the other. On a related note, what would be the benefit with regards to our hypotheses, when additionally providing estimates containing shared variance in addition to the current modeling of unique effects of each PE? From our understanding, these estimates would not be more valid or informative for testing our hypotheses which is why we would like to refrain from adding analyses to the manuscript with redundancy and little informative value.

Lines 687-688: "Parametric modulators were not orthogonalized, thus each only explaining their specific variance."

Comment #7d: * iv) I'm not convinced by the additional value of the third model. PE is a difference between feedback and performance expectation, and it does not help to show the separate effects of these components because they represent different psychological constructs. If you want to control for the feedback level (or performance expectation), then better include the feedback level as a first parameter and PE as a second, orthogonalized parameter within a parametric modulation of feedback, and report the effect of PE (see Mumford, 2015).

- We agree with the reviewer regarding the assessment of the additional analysis. In order to focus on our main hypotheses, we have now dropped this analysis from the manuscript. The rationale for initially running this analysis was to further support the PE processing in line with the prior analyses provided by (Zhang et al. 2020).

Comment #8: * Second-level GLMs: The whole paragraph (lines 677-689) is not easy to follow. It is not clear i) how many different models were specified, ii) what contrast images each of them included (PE/absolute PE or others), iii) why sometimes flexible factorial and sometimes t-test were used, iv) was only one contrast image for Self included in the "additional second-level models" (line 683) and only one covariate per model? v) Relating to the covariate-GLMs: Please specify that (according to Table S7) and why these analyses were restricted to ROIs.

- We apologize for the difficult read. To improve readability and in response to the reviewers comment, we described the second-level GLMs in more detail by stating the number of models and conditions included in each model. We also described the covariate models in more detail. Here, three separate models were used each including only the contrast for Self and one covariate. The reasoning behind using more complex models like flexible factorial or repeated measures ANOVAs was that we wanted to flexibly test different conditions against each other (e.g. Self > Other and Other > Self or the 2 x 2 interaction). T-tests were implemented for all baseline contrasts (e.g. Self vs baseline) because SPM does not allow testing for baseline contrasts in more

complex models if repeated measures are used because error variances are calculated incorrectly per default and effects would be falsely inflated. If it would have been possible with SPM, we would have preferred to only use one more complex model (or at least as few as possible) for all comparisons. We also specified in the methods section that ROI analyses were conducted and why.

Lines 698-704: "Four separate one sample t-tests were implemented for PE valence and PE surprise for Self and Other. For direct comparisons of PE valence and PE surprise responses for Self and Other, two repeated measures ANOVAs were conducted including the respective beta images for Self and Other. Differential tracking of the PE valence, depending on biased learning and self-conscious affect, were examined by three additional second-level models for the PE valence beta images for Self including either the Valence Learning Bias, embarrassment, or pride ratings as between-subject covariate."

Lines 710-713: "The analyses including all covariates were conducted within our predefined ROIs, the bilateral dAI, vAI, amygdala, mPFC, and VTA/SN, as these regions are associated with affective and motivational aspects and PE tracking during learning (for a detailed description see below)."

* PPI analysis:

Comment #9a: * i) What does it mean to "assess whether there was a stronger functional coupling of the dAI with our predefined ROIs ... when PE valence [pos arrow neg] was more negative"? Did the authors test this via the contrast -1 to identify connectivity that would be stronger the smaller PE was? Please explain why you focused on the negative effect specifically. Was the opposite contrast tested as well (stronger connectivity during larger, i.e. positive PE)?

- Yes, we set the contrast to -1 and tested for connectivity that was stronger for more negative PEs. The rationale behind it was that we have very strong evidence for greater learning from negative PEs compared to positive PEs and an overall negative learning bias with this task in our previous studies (see (L. Müller-Pinzler et al. 2019; Czekalla et al. 2020) and also in this study. This pattern in the LOOP-task suggests a preference for negative information when making updates on the own performance cognitive estimation, which should be related to neural activity and functional connectivity when receiving feedback; if negative PEs result in greater updates, we should also find greater connectivity during feedback presentation compared to cases of positive PEs. In this line, pupil dilation responses were also stronger with more negative PEs in contrast to positive PEs. Due to this specific hypothesis, we did not test the opposite effect. We added another sentence describing this in more detail.
- ***Lines 738-744: "In line with the negative Valence Learning Bias for self-efficacy beliefs and stronger pupil dilation responses scaling with more negative PEs we specifically tested for stronger functional connectivity correlated with more negative PEs."***

Comment #9b: * ii) Lines 715-718: Was this a separate GLM with Valence Learning Bias included as a covariate? Which predefined ROIs do you refer to here: dAI and/or others?

- The predefined ROIs we referred to are amygdala, mPFC, VTA/SN. Insula ROIs were excluded from the PPI analyses because the insula already served as the seed region. We added the list of ROIs in the description as suggested. Correlation analyses were conducted by extracting the average parameter estimates from the original PPI GLM from a sphere of 6mm around the peak voxels within the predefined ROIs (amygdala, mPFC, VTA/SN) and Spearman correlations were calculated with the Valence Learning Bias (see also Figure 4).

Lines 740-744: "Functional connectivity dynamics were also associated with learning behavior by calculating Spearman correlations for the Valence Learning Bias and the parameter estimates for the PPI effect of Self > Other extracted from the GLMs described above in a sphere of 6mm around the peak voxels within the predefined ROIs (amygdala, mPFC, VTA/SN)."

Comment #10: * ROIs: Please specify whether VTA/SN was an anatomically specified ROI and if yes based on which atlas.

- VTA/ SN are midbrain ROIs based on anatomical landmarks and derived from the probabilistic atlases of the midbrain by the Adcock Lab (Murty et al. 2014; Ballard et al. 2011). We added a more detailed description in the revised methods section.

Lines 754-758: "Additionally, an anatomically defined VTA/ SN ROI, dopaminergic nuclei in the midbrain, was included (probabilistic atlases of the midbrain; Adcock Lab) (Murty et al. 2014; Ballard et al. 2011) as dopamine signals motivationally important events, e.g. during reward learning (Schultz 1998), and has been associated with biases in memory towards events that are of motivational significance (Adcock et al. 2006)."

* fMRI significance threshold:

Comment #11a: * i) "FWE corrected on the whole brain level" (line 732-733, plus e.g., Table S5) is underspecified - you can apply the FWE correction at the cluster or peak level, and both can relate to the whole brain or to a small volume. Please specify more precisely.

- We thank the reviewer for this comment, FWE correction for the whole brain, and the ROI analyses was applied at peak level. We specified this throughout the whole manuscript.

Comment #11b: * ii) Please use the same statistical threshold for all brain analyses (e.g., FWE-corrected at the peak level + extent threshold, or FWE-corrected at the cluster level + cluster-defining threshold), or clearly justify why the threshold needs to be adjusted for the different analyses.

- We agree with the reviewer that we should use the same statistical threshold for all brain analyses. Our aim for using a more lenient threshold for the effects of PE surprise for Self and Other was that we wanted to show the overlap of activation for Self and Other. We now decided to show the overlap in Figure 3 by presenting the results uncorrected for displaying purposes and still providing all statistics at the same threshold as the other statistics (FWE corrected at peak level for the whole brain, see Table S4 and results section of the main manuscript). We therefore updated Table S4 and Figure 3 accordingly.

Comment #11c: * iii) Before applying the small volume correction for ROI analysis, which whole brain significance threshold was applied?

- Our procedure was to look for activity at the whole brain at $p < .05$ uncorrected. This, however, should not invalidate FWE correction for the ROI analyses as these were conducted on the peak level so that the inflated cluster extent should not have any impact.

****Results:****

Comment #12: * BMS: Please report the expected frequency in addition to p_{xp} in order to indicate the effect size.

- We added the expected model frequency for the winning model in the results section.

Line 177: "The expected model frequency was 46.53."

Comment #13: * Behavioral analyses: Please report effect sizes as well.

We have added the effect sizes where appropriate.

Comment #14: * fMRI results are not communicated clearly enough with respect to their technical description and their interpretation, please double check and correct through out the manuscript. Arbitrarily selected examples just to make the point clear (not a complete list):

Comment #14a: * i) Supplementary, lines 49-50: "Parameter estimates for the differences between feedback for positive vs. negative PEs derived from the peak voxels of the interaction effect depict the interaction in the left NAcc/VS ...".

Why not something like: "Parameter estimates correspond to the difference in BOLD response to positive relative to negative PEs in bilateral NAc [MNI coordinates]. The plot shows that positive relative to negative PEs increased the activity in NAc only when learning about the own performance, but not when observing others."

In addition, the plot title rather refers to the y-axis than to the general plot, that additionally differentiates between self and other. Furthermore, plotting 4 different contrast estimates (self-PE-pos, self-PE-neg, other-PE-pos, other-PE-neg) would provide a more complete information about the interaction effect (e.g., was there a deactivation in response to self-related negative PE and an activation in response to self-related positive PE?).

- In line with the reviewer's suggestion, we checked the fMRI results section and improved the descriptions throughout the manuscript. We also changed Figure S3b according to the suggestions, changed the title and plotted all four conditions.

Comment #14b: * ii) Table S4: "... associated with the parametric weights of ..." is a verbal double dipping. "Activation associated with (or correlated with) absolute PEs" would be enough. The anatomical label "Temporal Pole/*" seems to be displaced and to refer to the

cluster below (size 107). And what do you mean with "Cluster extends refer to ..."? Do you mean "cluster extents"? Still, this is an unusual wording to specify the significance level, and makes only sense for FWE-correction at the cluster level (e.g. same wording is used in Table S5, with the additional note that "Only clusters with more than 50 voxels are reported...").

- We checked our tables and changed the wording according to the reviewer's suggestion. The sentence "Only clusters with more than 50 voxels are reported" does not refer to a cluster level correction, it only refers to the presentation of the peak level FWE corrected results in the tables. In some tables we did not report all significant peaks when there were less than 50 voxels in one "cluster" surviving peak level correction because some tables would have been very extensive with several single voxels spread throughout the brain surviving peak level correction.

Comment #14c: * iii) Table S7: "Covariates Associated with Individual Differences ..." again verbal double dipping. "Covariates revealing individual differences..." for instance sounds better.

- We changed the wording to variables to avoid verbal double dipping.

Comment #14d: * iv) I would suggest not to use the quite unusual label "PE valence [neg arrow pos]" to describe the parametric modulation of the BOLD response to feedback by PE. This is confusing, as (here or in other analyses) the authors simply refer to (signed) prediction errors, which is a parameter well established and often reported in the literature. To make things worse, the authors also use "PE Valence" (Table S5) and "Valence Specific Prediction Error Tracking" (Table S7) to refer to the same effect, and "PE valence [pos | neg]" to refer to a different, categorical variable. Wouldn't it be easier to simply refer to PE, absolute PE ("PE surprise"), and PE valence (categorical variable, as valence has only two levels, positive and negative) to differentiate these effects?

- We agree with the reviewer that our choice of labels might have been somewhat confusing. We however slightly disagree that valence has only two

levels and cannot be conceptualized with a continuum. The concept of valence as a continuum is very much rooted in the theories of emotion and validated across many different studies which, besides the arousal that stimuli elicit, differentiate the valence from very negative to very positive. We have nonetheless discussed this issue in greater detail, to avoid confusion with our labeling and we think we have found a solution that should work better. Our reasoning for choosing “PE valence” as a label instead of a more neutral description like “PE” was that “PE valence” already transports information about the meaning of this variable of interest. In our opinion other descriptions such as “signed PE” are rather technical and need additional explanation to understand these in terms of psychological constructs and their affective meaning (i.e. more negative PEs having a more negative valence and more positive PEs having a more positive valence). We, therefore, prefer to keep “PE valence” as a label for the continuous effect of signed PEs.

To resolve the conflict with our labeling we chose an alternate label for the categorical effect of the learning rates for positive vs. negative PEs as derived from the computational model. Since the categorical effect compares the learning rates for positive and negative PEs we now label the factor in the model “Prediction Error Sign”. This emphasizes the categorical differentiation (positive vs. negative prediction errors) instead of the continuum associated with the valence of the prediction error. Overall, we feel that with this labeling misunderstandings should be ruled out for the two occurrences of the factor label in the result section and the additional sections that have been moved to supplementary information (see also the changed Table S7). With the remaining differentiation of “PE Valence” and “PE Surprise” we hope to have now clarified the meaning and use of the different variables for the reader and found a solution that carries the relevant information.

Lines 192-197: "There was also a main effect of Prediction Error Sign ($F(1,67)=5.22, p=.025, \eta^2=0.011, \text{partial } \eta^2=0.072$); categorical comparison of

learning rates for positive vs. negative PEs) and a significant interaction of Agent x Prediction Error Sign ($F(1,67)=21.47$, $p<.001$, $\eta^2=0.040$, partial $\eta^2=0.243$), which replicates earlier findings of a bias towards more negative updating when learning about one's own performance ($t(134)=-4.85$, $p<.001$, $t(68)=-3.53$, $p<.001$, $d=-0[3] .425$, $M\alpha_{Self/PE+}=0.25$, $SD=0.13$; $M\alpha_{Self/PE-}=0.35$, $SD=0.20$ (Müller-Pinzler et al. 2019))."

Lines 612-616: "A repeated measures ANOVA was calculated on the learning rates with the factor Agent (Self [$\alpha_{Self/PE+}$, $\alpha_{Self/PE-}$] vs. Other [$\alpha_{Other/PE+}$, $\alpha_{Other/PE-}$]) and factor Prediction Error Sign (PE+ [$\alpha_{Self/PE+}$, $\alpha_{Other/PE+}$] vs. PE- [$\alpha_{Self/PE-}$, $\alpha_{Other/PE-}$])."

Lines 713-717: "We additionally performed psychophysiological interaction (PPI) analyses on the first level, investigating whether functional connectivity of the dAI, which is commonly activated during feedback processing independent of Agent and Prediction Error Sign (conjunction of baseline contrasts: feedback Self ^ feedback Other), would differ depending on the PE valence."

Comment #14e: * v) Be careful when saying that the activity increased in response to negative vs. positive PEs (e.g., line 290) when referring to the parametric modulation of the BOLD response to feedback by PE. "Negative vs. positive" sounds like a categorical variable (disregarding the size of the PE, and taking into account only its valence), but you actually refer to a correlation between 2 parameters. Therefore, you need to say that the activity scaled with increasing (decreasing) PE, or positively (negatively) correlated with PE, etc.

- We agree with the reviewer that our wording was misleading. We changed our wording throughout the manuscript in line with the suggestions.

****Discussion**:**

Comment #15: * Lines 403-404: Why is the "common currency" assumption related to absolute PE ("PE surprise")? Cited Ruff and Fehr (2014) and other relevant publications

specifically refer to value computation: "neural value signals that encode the rewarding properties of the choice options". This is related to the signed PE, where large positive numbers reflect positive values and small negative numbers reflect negative values. Instead, large absolute PE reflect both positive and negative values and therefore cannot be validly related to theories and findings about value computation.

- We agree with the reviewer that the concept of "common currency" does not really fit our idea of a common process for tracking information of negative and positive value. We, therefore, removed the corresponding section from the revised version of the discussion.

Comment #16: * Lines 409-411: What exactly do you refer to: The modulatory effect of absolute PE and the one of PE? This comparison is not valid as the two parameters relate to different psychological constructs, and not merely to different extents of balance.

We agree with the reviewer that this section is unclear. To clarify, we did not intend to compare the effects of both the signed and unsigned PE but aimed to explain their additive effects on the BOLD response in the same brain regions, as these effects overlay each other in their modulation of the HRF. More precisely, we expected greater unsigned PE (i.e. surprise) to be associated with stronger BOLD responses in regions associated with error tracking, such as the AI and mPFC. Accordingly, this effect is robust across individuals, regardless of their affective experience and learning biases. In contrast to this, valence-specific modulation of BOLD activation in those regions is associated with learning biases and affective experience but is not generalizable across subjects regardless of emotional experiences and learning. The sum of both effects indicates that the U-shaped error-related modulation of the BOLD response in these regions is unbalanced in accordance with affective experiences and learning: in the case of more negative affect and with a greater negativity bias, the response toward negative PEs is more pronounced, compared to positive PEs.

We have changed the section accordingly to better explain the summarized shape in the modulation of the BOLD response in these regions.

Lines 418-430: "The tracking of the absolute error, PE surprise (Rouhani and Niv 2021), independently of the agent, suggests that there is a common and valence-independent coding of surprise in the insula and the mPFC that could be sufficient to complete the learning task per se. As our results indicate, however, the valence of the PE is relevant for understanding the trajectories of how individuals form self-efficacy beliefs. This is implicated by a valence-dependent, additive shift in the error-related BOLD response of these regions that corresponds to individual differences in the learning bias and affective experience. As a result, besides the main effect of surprise on the BOLD response of the AI and the mPFC, individuals who form more negatively biased self-efficacy beliefs and experience more negative affect, also have greater error-related responses in the case of negative PEs in contrast to positive PEs. This congruency in the modulation of the U-shaped surprise function hints at a neurocomputational mechanism of how affect may shape the formation of beliefs, as proposed previously (Bromberg-Martin and Sharot 2020)."

Comment #16: * Could you shortly comment on why you think people rely more strongly on negative PEs when learning about their efficacy, but do the opposite when for example estimating future risks as reported in the "optimism bias" literature?

- Thanks for this remark! In fact, this question cannot be fully answered with the current literature, but clearly is a question that we will follow up on in the future. It is an interesting question of why sometimes and under certain conditions people exhibit a positivity bias and sometimes a negativity bias.

We assume that in situations that contain a potential future risk the positivity bias may regulate mood. In situations that contain something that is likable, but not changeable, such as e.g. learning about one's intelligence, a positivity bias would strengthen the participant's self-view and enhance self-worth.

Moreover, in both situations, the participant can not even potentially “improve” or “work” on the outcomes as all trials are isolated and unbound inquiries relating to different risks or intelligence parameters. In our task, learning (processing the feedback) in one trial may be informative for a future trial - thus participants might be more inclined to focus on trials where the PE is negative as these are the trials where improvement is necessary (to focus on what is most needed). Given the participants feel they have the necessary psychological resources (i.e. sufficient self-esteem, low self-threat induced by a novel task), we think it makes sense to focus on the negative feedback. Another argument could be that tasks on intelligence and health issues are not normatively neutral. This could be an advantage of our task in which participants have rather low prior assumptions. Our participants do not have strong assumptions about their capability in the cognitive estimation of weights of animals. Other arguments for a more negative learning bias might be that participants have more negative prior assumptions about their estimation ability compared to other abilities and estimating is also less central to their self-concept which makes a stable positive ability concept less necessary to feel positive about oneself.

We have discussed this issue in our behavioral paper before (L. Müller-Pinzler et al. 2019), but are currently working on a more elaborate perspectives article on the conditions that favor either a positivity or negativity bias.

****Minor comments**:**

Comment #18: * Was it on purpose that the word "affected" in the title is open to interpretation? It could mean both affected in the sense of influenced by something, and affected in the sense of affective or emotional. If not, "affective" instead of "affected" might be more straight forward. While the term is explicitly introduced in the manuscript (line 73, however not in the abstract), the title may be difficult to understand without this explanation.

- In fact, it was exactly the idea of this title to point to the twofold meaning of the word “affected”. As it is central to our argument in this paper that learning

of self-efficacy beliefs potentially is “influenced” in two ways: first by biases (negativity bias as potentially being indicative of the motivation to improve during the course of the performance learning task) and second by the emotionality experienced in the context of the learning task (embarrassment & pride as central social emotions that only are formed during successes and failures when accomplishing a performance task that can be attributed to one’s own (un)ability). Thus, we would prefer to keep the title, but be more precise in the abstract and the introduction when we refer to “affected beliefs”.

- Accordingly, we state the following in the introduction:

Lines 66-69: “It is proposed that this internal affective information is integrated with external information to shape beliefs that rather than being objective, are motivated and biased by subjective feelings about the beliefs themselves, leading to a recursive influence of beliefs and affective states on each other (Loewenstein 2006; Bromberg-Martin and Sharot 2020; Kunda 1990).”

In the abstract, we hinted toward the double connotation of the word “affected” in the second sentence but understand that the link is not obvious enough. We have thus rephrased the abstract to make it clearer that we refer to two kinds of “influences”: bias and affect.

Comment #19: * line 64-66: another relevant reference could be Kunda, 1990, The Case for Motivated Reasoning (optional)

- Thanks for pointing us to this missing citation! It is now added.

Comment #20: * line 69 and others: another relevant reference could be Vinckier et al., 2018, Neuro-computational account of how mood fluctuations arise and affect decision making

- Thanks also for pointing us to this and related publications, which we were unfortunately not aware of. One reason could be that in these papers the main variable was “fluctuations in mood” and there was no explicit mention of the terms “emotion” or “affect”. Anyway, we think that (Rutledge et al. 2014, 2016; Stolz et al. 2020; Laura Müller-Pinzler et al. 2015; Cecchi et al. 2022; Vinckier et al. 2019, 2018) and (Rutledge et al. 2014, 2016; Stolz et al. 2020; Laura Müller-Pinzler et al. 2015; Cecchi et al. 2022; Vinckier et al. 2019, 2018) are good examples of how mood/affective states impact (risky) decision-making.

We now write the following in lines 69-71: “Previous studies supported aspects of Bromberg-Martin & Sharot’s framework by demonstrating that internal beliefs and external feedback can elicit emotions like happiness, pride, or embarrassment (Rutledge et al. 2014, 2016; Stolz et al. 2020; Laura Müller-Pinzler et al. 2015; Cecchi et al. 2022; Vinckier et al. 2019, 2018)”

Comment #21: * What do you mean with "average number" (e.g., line 492)?

- By “average number” we referred to the mean frequency of negative or positive PEs during the task per participant. We changed the wording accordingly.

Lines 506-515: “This procedure led to varying feedback sequences between participants but kept PEs mostly independent of the participants’ performance expectations and ensured a relatively equal distribution of negative and positive PEs across conditions (Self: mean positive PE = 13.6, SD = 1.8 (mean frequency = 20.3); mean negative PE = -12.6, SD = 1.4 (mean frequency = 19.7); Other: mean positive PE = 13.0, SD = 1.3 (mean frequency = 19); mean negative PE = -13.1, SD = 1.1 (mean frequency = 21)). At the beginning of each trial, a cue was presented indicating the estimation category (e.g. “height”) and the agent whose turn it was (e.g. “you” or “Tim”). Afterwards participants were asked to state their expected

performance for this trial on a scale with the same percentiles used for feedback."

Comment #22: * Figure 1: The legend says that the cue indicated both the estimation category and the agent, but the figure and the methods indicate that it was only the category.

- We thank the reviewer for pointing out this inconsistency. Apparently, we forgot to mention the agent in the cue in the methods section and did not depict it in the Figure either. We now changed both in the revised version. The Figure per se does not show the task with all the details exactly how it was presented to the participants but rather a symbolized version of the task presenting the most important aspects with e.g. much bigger font size. We, however, seemed to have missed an important aspect here and added it now.

Comment #23: Furthermore, why was the format of performance percentile scale changed from the expectation rating (white line) to the feedback (colored bar) instead of using one format for both scales? The method text says that it was the same scale, but the screen shots in the figure indicate a changed format.

- We chose the colored vertical scale during feedback because in our opinion it most intuitively shows the valence of the feedback (i.e. green is good and red is worse; high is good while lower is worse). During the rating phase, however, it is much easier, especially with restricted movement in the MRI, and also more intuitively to navigate the computer mouse on a horizontal line. To make the task easier to handle for the participants we presented all of our ratings in the same way: via a horizontal white line (emotion ratings, estimations, expectation ratings). This is the reason why scales do not match between rating and feedback. By stating that we used the same scale for rating and feedback we referred to the anchors of the scale from "better than 0%" to "better than 100 % of the comparison group". We now changed our wording to make this clearer.

Lines 513-515: "Afterwards participants were asked to state their expected performance for this trial on a scale with the same percentiles used for feedback."

Comment #24: * Please report whether the group mask in 2nd level fMRI analyses covered the whole brain or whether some areas (e.g., vmPFC and ventral striatum) were not completely included due to susceptibility artifacts, which may be more pronounced in multiband MR sequences.

- The reviewer raises an important remark. Brain coverage was rather typical as observed in fMRI studies with signal loss due to susceptibility artifacts in the OFC and close to the VS. However, as 2nd level masks indicated, coverage was acceptable for the targeted regions across individuals included in the present study. Most significant losses were observed in the OFC but we found preserved fMRI in the ventral striatum after preprocessing.

Figure R2. Brain mask of the 2nd level fMRI analyses. Brain coverage and signal loss was as expected for fMRI studies with preserved signal in our regions of interest.

Comment #25: * Please use inconsistent abbreviations across the manuscript, e.g., correct inconsistent condition abbreviations in lines 606-607 vs 612-613.

- We changed the abbreviations consistently.

Comment #26: * Maybe I missed something, but just to be sure (line 361-362): Have you reported the final efficiency beliefs as well (or the total change from the initial to the final belief, across the entire experiment)?

- We now added Table S9 with the initial and final beliefs for all Ability conditions to the Supplementary Information.

Comment #27: * Lines 386-388: What do you mean with idiosyncratic response patterns?

- We changed the wording of this sentence. Our aim was to point out that other studies show as well that there are rather large interindividual differences in the VTA response during reward learning tasks.

Lines 403-405: "The subjective value of self-related information varies significantly between subjects, as indicated by response patterns of the VTA/SN to gains or losses (Charpentier, Bromberg-Martin, and Sharot 2018)."

Comment #28: * When discussing the functional connectivity of the dAI, can you also provide some evidence on underlying anatomical connectivities that would support the assumed interpretation?

- Thanks for this remark. We have now referred to a classic paper showing anatomical connectivity between the insula with nearly all cortical association regions, the thalamus, and the medial temporal lobe, amygdala, and basal ganglia (Mesulam and Mufson 1982). Also, we have added two citations suggesting that the dorsal AI is basic or integrative functional role (Kurth et al. 2010; Kelly et al. 2012).

Lines 397-400: "The functional connectivity dynamics of the dAI, matching the modeled learning rates with a stronger impact of self-related negative PEs, underline the insula's role as an integrative hub (Mesulam and Mufson 1982) that receives and forwards signals affecting information processing in other brain regions (Kurth et al. 2010; Kelly et al. 2012)."

Reviewer #2 (Remarks to the Author):

In this manuscript, Müller-Pinzler and colleagues use a belief updating task in conjunction with self-report ratings of affect, computational modelling, pupil dilation and fMRI to characterise information integration in response to differentially valenced (negative, positive) information. Behaviourally they show that beliefs about oneself are updated asymmetrically with greater updating in response to negative vs positive information. This is nicely unpacked using computational modelling by fitting separate learning rates for positive ($\alpha+$) and negative ($\alpha-$), revealing a difference in the size of these rates ($\alpha- > \alpha+$). This learning bias is shown to correlate with subjective self-reports of positive (pride) and negative (embarrassment) with this bias (towards negative over positive learning) being enhanced with higher embarrassment and reduced for higher pride ratings. Pupil dilation during information presentation was higher for negative compared to positive PEs (controlling for unsigned PEs, i.e. general surprise/saliency). fMRI suggested that the mPFC responded to the general salience of the information (unsigned PEs) independent of valence and the self/other frame. Interestingly, responses in the NAcc, a region well known to respond to rewards such as money, were much more selective - responding to signed PEs specifically for self relevant information, consistent with the idea that information itself can be rewarding and is more rewarding the better than expected it is. The authors also examine how the parametric effect of signed PEs interact with a number of individual difference measures (pupil dilation, learning rates, subjective ratings) in a number of ROIs (amygdala, insula, MPFC and VTA/SN).

I am convinced by the behavioural results and computational modelling which replicates previous work from this group using this task (2019). However, I was a bit less convinced by the some of the results that followed this which made great efforts to try and link this interesting behavioural finding (asymmetry in updating) to self report, pupil dilation and fMRI measures. I detail these concerns below.

Comment #1: Ratings

Why did the authors choose to get participants to rate “embarrassment” and “pride” specifically? It seems the goal was to tap into a subjective sense of participants positive and negative affect, so why narrow it down to asking about these emotions specifically? Would it not have been easier to ask how happy/positive (vs unhappy/negative) the participants was? In other words, how much do the authors think participants are submitting ratings about these feelings specifically; would they expect the same results if they rated joyous/disappointed/satisfied/unsatisfied etc.

We thank the reviewer for this remark as it raises the relevant question of how to best capture the variability of interest in the affective experience in our task. It matters because with different affective anchors, we might measure our construct less validly. The reason for why we have decided to specifically ask for these emotional anchors while reporting on the emotional experience during learning, is that emotional theory suggests that pride and embarrassment are the most distinctive affective concepts that people use to construct and communicate emotional experience in such a context. In contrast to happiness, theory of self-conscious emotions understand pride and embarrassment to belong to unique emotional construals that are, among other things, distinguished from other emotional concepts by essential self-related appraisals and activation of self-concepts (see (Tangney, Stuewig, and Mashek 2007)). When it comes to emotional experiences in context of performance situations, pride or embarrassment are thus theoretically more valid to capture relevant differences in affect. We are aware of many studies successfully using rather core affective dimension on the valence continuum (from unhappy to happy) when assessing learning biases and in the context computational modeling (see (Rutledge et al. 2014)), however, our own previous research has also experimentally demonstrated that when people experience internal control and can attribute outcomes to the own performance, assessment of self-conscious emotions better reflect the

affective responses (see (Stolz et al. 2020). For example, we did find robust evidence for greater differences in pride compared to happiness when people experience internal control in a performance situation (i.e. were able to attribute task outcomes on their own performance) and embarrassment to be more sensitive in to failure during cognitive estimation in a set of other emotional, more general assessment of affects (see (Laura Müller-Pinzler et al. 2015). In summary, these findings made us choose pride and embarrassment as core emotional anchors for assessing the overall affective experience in our task, as they should measure emotional experiences with greater validity and show stronger differences between individuals in performance situations.

Thus, we have now added two sentences to support our decision to ask for experiences of pride and embarrassment:

Lines 104-109: “Self-conscious emotions differ from other emotional concepts as they essentially involve self-referential evaluations and the activation of self-concepts (Tangney, Stuewig, and Mashek 2007). Thus, when it comes to emotional experiences in the context of a performance situation, pride or embarrassment are theoretically more valid constructs to capture differences in affective experiences than e.g., the basic emotion happiness.”

Notably, this does not imply that the experience of pride and embarrassment are unrelated to the experience of (un)happiness, joy, or satisfaction. Emotion theory clearly suggests that these emotions have comparable underlying valence, should be correlated with another, and can also be expressed for communicating emotional experiences to the same event (Scherer 2005). People hardly report and feel a unique emotional expression, but they can use many different words for informing others about their internal state. When people experience pride, they will also experience joy, happiness, and satisfaction and when they are embarrassed, they also report to be unhappy, stressed, or disappointed. Accordingly, we found the expected associations in

previous studies (Stolz et al. 2020), for the association of happiness and pride, and (Laura Müller-Pinzler et al. 2015) for the association between embarrassment and other negative emotional concepts) and we would not be surprised if effects would be comparable if we would have used the suggested anchors. However, since we would be measuring affective constructs that put less emphasis on self-related appraisals we expect these associations to be weaker for the LOOP task as this task requires learning about the self.

We have nonetheless tested this notion by additionally correlating learning biases with other, more general affective constructs that have been part of our assessment. We have also asked subjects how much tension (German: "Anspannung") and how happy (German: "Freude") they feel. Correlations of affective states with another were as expected with happiness being most positively correlated with pride ($\rho = .39$) and tension being most positively correlated with embarrassment ($\rho = .53$). When it comes to associations with the learning bias, correlations are in a similar direction but smaller compared to the ratings of pride or embarrassment. Tension was not significantly associated with the learning bias ($\rho = -.08$), happiness was only trendwise associated with the bias ($\rho = .23$, $p = .059$, see the table for reference below) but less so than pride ($\rho = .55$, $p < .001$). These results support our rationale and expectations above. To provide readers a more comprehensive overview about the correlations of learning rates with affective experience in our task, we have added this table to our supplementary material (Table S10).

Spearman Correlations

	Valence Learning Bias	Embarrassment	Pride	Happiness	Tension
Valence Learning Bias		-0.24*	0.55**	0.23	-0.08
Embarrassment	-0.24*		-0.10	-0.07	0.53**
Pride	0.55**	-0.10		0.39**	0.20
Happiness	0.23	-0.07	0.39**		-0.07
Tension	-0.08	0.53**	0.20	-0.07	

Table S10. Spearman correlations for all emotion ratings and the Valence Learning Bias. * p<.05; ** p<.01

Comment #2: I was not clear what emotions participants rated for each condition. Very simply, could the authors state for each of the 4 conditions (high ability self, low ability self, high ability other, low ability other) which emotions (Embarrassment, Pride, Happiness, Stress/Arousal) participants were asked to rate, how many times they rated each of these emotions and when they rated them (e.g., start/middle/end of a condition). If specific emotions (e.g., Pride) were probed for specific conditions (e.g., high ability self) is there a risk that this could cue participants about the conditions?

- We thank the reviewer for bringing up this point. We asked participants to rate emotions after a finished trial, this means following the pause after the

feedback phase, and during every emotion rating phase all four emotions were prompted. Thus, in four emotion rating phases all emotions were rated once for each of the four conditions (high ability self, low ability self, high ability other, low ability other). We therefore think that there is no specific association between emotions and conditions and emotions ratings also did not serve as a cue for a specific condition. We added a more detailed description in the methods section.

Lines 534-536: "During the whole task four emotion rating phase, including all four emotions, were presented, each following a trial of one of the four experimental conditions (e.g. Self - High Ability)."

Comment #3: If participants only rate each emotion once (or twice), could it be that the emotions being rated are not anything to do with the task and the information participants are getting during it but is much more general (i.e. how they generally feel).

- We agree with the reviewer that this question is always difficult to answer. We believe that the emotions ratings reflect both, a response to the task but also have a component that is specific for the individual. To support this assumption we looked at the correlations between the two emotion ratings for Self - High Ability and Self - Low Ability, which are rather high for embarrassment ($\rho=.76$) and pride ($\rho=.62$) showing that there is a rather stable difference between subjects. Aggregating emotions across both conditions therefore makes sense. However, when correlating embarrassment ratings with a questionnaire for susceptibility to experience embarrassment in daily life (susceptibility for embarrassment scale), correlations are much lower (Self - High Ability: $\rho=.18$, Self - Low Ability: $\rho=.04$) indicating that ratings might not simply reflect how participants generally feel or how they assess their personality. When we compare the emotion ratings between the two self-related conditions we also see that at least pride ratings differ with respect to the feedback presented before (Self - High Ability: mean=47.9, standard error=2.3, Self - Low Ability: mean=37.0, standard error=2.9). We did not find the same response for

embarrassment (Self - High Ability: mean=22.7, standard error=2.7, Self - Low Ability: mean=22.0, standard error=2.7) but we also did not specifically ask participants to refer their emotion ratings to the trial before. In a previous study we asked participants to rate their emotions specifically with respect to the feedback they received and found more pronounced differences for embarrassment as well (Laura Müller-Pinzler et al. 2015). Overall, we believe that these correlations show that the emotion ratings in our task are (i) task specific rather than habitual feeling states, are (ii) rather weekly driven by trial specific outcomes and reflect a sustained experience during the task, and (iii) can thus be aggregated across the assessments after different outcomes. These aggregated emotion ratings thus indicate a measure for an individual specific feeling state elicited during the task. However, we agree that it could be very interesting for future studies to assess emotion ratings in more detail and look at trial specific ratings as well (considering all side effects such a procedure would bring up for the fMRI design).

Comment #4: Pupil dilation

I am not an expert in pupil dilation analysis. But with that caveat, some potential issues that occurred to me are:

Comment #4a: (1) The task is not optimised to look at pupil dilation. Did the authors control for things like luminance, distance from the monitor, etc.

- We thank the reviewer for pointing out that this aspect has not been described in detail. The task has however been optimized for the analyses of pupil dilation with respect to several aspects and we have paid significant attention to not present results that might be confounded. The distance of the eye tracking system from the monitor has been controlled between subjects and should not lead to systematic distortions for different screen positions. We also controlled the global luminance of the stimuli, as stimuli were neither very light nor dark leaving room for further pupil dilation and constriction in the relevant task periods. The feedback screen has also been controlled for local

luminance, which means that all areas of the feedback scale have similar luminance making positive and negative feedback comparable. It would have been problematic, if subjects e.g. receiving more negative feedback, also fixate more frequently on aspects that have lower luminance. We however did not show the respective feedback scale in our current figures, as they have been designed for illustrative purposes. We have adjusted these figures and we added a more detailed description of this task in the methods section. The difference between self and other related feedback could not be controlled with respect to the effort participants invest into the task, which impacts pupil dilation as well. We therefore decided to only analyze self-related feedback and did not compare self vs other in our analyses as any results might be confounded by offsets at the beginning of stimulus presentation (see also Comment #6 of Reviewer #1).

Lines 631-636: "Stimuli were presented on a TFT display (32", Active area 698.4mm(H) x 392.85mm(V); Pixels 1920 x 1080; NordicNeuroLab's LCD monitor (NNL MRI InroomViewingDevice; NordicNeuroLab AS, Møllendalsveien 65 C, 5009 Bergen, Norway), located 50 cm from the observer, in an otherwise dark room. The task has been optimized with respect to eye-tracking by controlling the global luminance of the stimuli as well as the local luminance of the feedback scale."

Comment #4b: (2) The authors regress mean pupil dilation during feedback on each trial against signed and unsigned PEs (both in the same model) in a mixed effects model. Even though the two regressors will be completely uncorrelated (as half will be perfectly positively correlated and the other perfectly negatively uncorrelated rendering the overall correlation 0), this seems like it will create an odd tiling of the space whereby you have no instances in which unsigned PEs ("surprise") are high and signed PEs ("valence") are low and also no instances where unsigned PEs ("surprise") are low and signed PEs ("valence") are high. Is that right? Have the authors done any simulations to guard against the possibility that this might create some odd effects.

Yes, it is correct that the sizes of signed and unsigned PEs always matches, i.e. there are no instances, by definition, in which a large signed PE co-occurs with a small unsigned PE, or vice versa. We would like to point out that such a situation would in fact be rather artificial, given the tight mathematical relation between signed and unsigned PEs. In addition, on a conceptual level it appears implausible that a given feedback could indicate that one is much better (or worse) than expected, without simultaneously eliciting a corresponding surprise. For these reasons we had not initially performed any simulations regarding a scenario in which the tiling of signed and unsigned PEs is less odd or even completely balanced.

However, we do agree that aside from these considerations it should be made sure that the “odd tiling” does not affect the fixed effects (FFX) estimates obtained by fitting our mixed models. To tackle this question, we simulated 200 sets of new pupil dilation data for each subject for five different degrees of “oddness of tiling” of signed and unsigned PEs (see below). We then obtained FFX estimates from each of these 200 sets of simulated data to test whether the distributions of FFX estimates obtained by our mixed models were affected by tiling. For both data simulation and model estimation we used the same underlying model as used to analyze the empirical data reported in the manuscript. Some key aspects of this procedure are the following:

1. To assure that our simulated data were similar to the empirical data, all simulations were based on the empirical data and the distributions of effect estimates (FFX and RFX) obtained from these data. E.g., the noise component in the simulated data was generated by drawing from a normal distribution with zero mean and a standard deviation equal to the residuals from our empirical models. Similarly, RFX intercepts were drawn from a normal distribution with mean and standard deviation taken from the RFX intercepts obtained by our empirical model.

2. We were mostly interested to test whether the effects involving signed PEs (i.e., their fixed effect and the cross-level-interaction with Valence Learning Bias for self-related information) were unaffected by tiling. Therefore, to induce different degrees of tiling, we only changed the data for unsigned PEs before generating a new set of simulated data. To induce different degrees of tiling, we shuffled a subset of the empirical unsigned PEs on each simulation run. The subset was randomly selected from all 1440 data points on each simulation and different percentages of these data points were shuffled (0%, 25%, 50%, 75%, or 100%). Here, 0% maintained the odd tiling as present in our empirical data, whereas when 100% of data points were permuted, the tiling of signed and unsigned PEs was completely “non-odd”, i.e. balanced. 25%, 50%, and 75% permutations thus created intermediate degrees of tiling oddness.
3. The shuffling of unsigned PEs was the only thing changed in the predictors used to simulate new data. I.e., both the Valence Learning Biases (second level predictor) and the signed PEs remained untouched and thus identical to the empirical data.
4. On each simulation run, we created a single set of new regression weights for each participant which was then combined with each of the five levels of tiling to compute new data based on the first-level regression formulas (This is important, because thereby we assured that potential differences between tilings could not be attributed to random differences in the regression weights). We then added noise to these data (see above) and aggregated all subjects’ simulated data in a new data frame, based on which the mixed effects models were then estimated.
5. These simulated datasets were then analyzed with the same mixed model used in the manuscript, to recover FFX estimates for the parameters underlying each set of simulated data.

First, we verified that the FFX recovered from simulated data were in a domain of parameter space that, as intended, coheres with our empirical FFX. To this end, we plotted our empirical FFX estimates (as red dots) overlaid on top of violin plots of the recovered estimates in Figure R3. In this figure, we additionally displayed the means of the recovered parameters as empty black circles. This plot suggests that the recovered parameters did not deviate from the empirical ones for any level of tiling. This was corroborated formally by t-tests showing no significant differences between empirical and recovered FFX (all $p > .07$, uncorrected). Moreover, no differences between levels of tiling were found for directed (all $p_s > .154$) or absolute deviations (all $p_s > .184$) of recovered from empirical FFX. Together, this demonstrates that recovered FFX were in realistic domains of parameter space, and independent of tiling levels.

Figure R3. Empirical FFX estimates (as red dots) overlaid on top of violin plots of the recovered estimates; the means of the recovered parameters are displayed as empty black circles

Second, we tested whether the FFX used for data simulation were positively correlated with the recovered FFX, as would be expected if parameter recovery was successful. To do so, we tested correlations between each set of recovered FFX (e.g., the intercepts) and the corresponding parameter underlying the simulated data. Here, all correlations were positive (all $p < .032$, Holm-corrected), with the only exception being the FFX for unsigned PEs in a single level of shuffling (100%; completely even tiling), which just missed significance with $p = .053$. Together, this further corroborates that FFX recovery was successful and not systematically affected by differences in tiling (see Figure R4).

Figure R4. Correlation plots between each set of recovered FFX and the corresponding parameter underlying the simulated data.

In a final step, we tested whether the degree of tiling affected the quality of FFX recovery. For each fixed effect (intercept, PE, LRbias, absolute PE, LRbias * PE) we tested whether levels of tiling induced over- or underestimations of FFX (as compared to the parameters underlying the simulated data) by using the differences between recovered and underlying parameters as the dependent variable. No such effect was found for any of the five FFX (all $p > .153$; see Figure R5, top). In a similar fashion, we tested whether the precision of FFX recovery was affected by levels of tiling by performing equivalent analyses of variance for the absolute differences between recovered and underlying parameters. Again, no such effect was found for any parameter (all $p > .194$; see Figure R5, bottom).

Figure R5. TOP: Box plots for the differences between recovered and underlying parameters (FFX estimation error) for each fixed effect (intercept, PE, LRbias, absolute PE, LRbias * PE) and all levels of tiling. BOTTOM: Box plots for the absolute differences between recovered and underlying parameters (absolute FFX estimation error) for each fixed effect (intercept, PE, LRbias, absolute PE, LRbias * PE) and all levels of tiling.

Together, our analyses demonstrate that differences in tiling of signed and unsigned PEs did not induce systematic biases during data simulation or the estimation of FFX from these simulated data. We are thus confident that our analyses of empirical data were equally unaffected by such undesirable effects.

Comment #4c: (3) Why did the authors only include the signed PEs (and intercepts) as random (per subject) effects but not the absolute PEs (which are only in there as fixed effects)? I could not see any rationale for this choice of model but maybe I missed it.

- We agree with the reviewer. In line with the suggestion we included random effects for the absolute PEs, too, and updated the results and methods section accordingly. The results did not change substantially.

Comment #4d: (4) A “classic” pupil dilation response peaks around 2 seconds after onset (e.g., of information). In the past I’ve seen researchers take this peak (or peak minus dilation at trial onset) as their trial by trial measure of dilation (rather than taking the average the entire response over the trial. But the pupil response in the authors data looks rather different to this (orange trace in Figure 2b) – why is that the case? And what is the rationale for averaging responses over the entire window (e.g., versus taking the peak, peak minus trough, etc.)?

The reviewer raises a very important remark on how to best summarize the pupil’s signal during feedback processing. In fact, the dynamics in the pupil reflect different processes at different onsets and temporal expansion and we

want to more deeply elaborate our rationale for our procedures below. We are not sure what the “classic” pupil response would look like but for the purpose of our study two different signals should be relevant to distinguish, one of which the reviewer is referring to above.

First, since PE valence is our target predictor variable we were aiming to capture variance in the pupil dynamics that reflect arousal and autonomic activation. In line with classic studies (e.g. Bradley et al. 2008) or following analyses on the properties of various physiological signal in response to stimulus lengths (e.g. (Geuter et al. 2014) greater pupil dilation in response to autonomic activation and emotional arousal begins to build up between 1.5 and 2 seconds after stimulus onset and after that persists and increases to differentiate between high and low arousing states with longer stimulus duration. The latter is important as longer stimulus durations and more negative valence lead to a sustained increase in pupil size for the period of stimulus presentation (see e.g. Geuter et al. demonstrating the persistence of the differentiating potential also for stimuli of very long duration). We agree that summarizing this signal with the slope of change in pupil diameter might not appear to be a frequent choice in context of studies using pupil dilation as a physiological response (see our second paragraph below). Based on these considerations. the question should however not be focused on whether the period of 1-2 seconds after stimulus onset would better capture our signal of interest, but whether we should have better focused on the last e.g. 500 ms of the stimulus presentation as this signal builds up over time. We decided to take the slope as it does conveniently capture the relevant pupil dynamics for a complete trial, without further specifications of a time window of interest at the end of the trial and it does very well reflect differences in arousal in case stimuli are presented for around 3-4 seconds as indicated by earlier studies. For example, such prolonged differences in pupil size could photographs of another’s injuries from neutral pictures (Paulus et al. 2015), was related to subjective experience of embarrassment or vicarious pain (Paulus et al. 2015; Laura Müller-Pinzler et al. 2015), and also trial-by-trial variation in BOLD signal

during affective stimulus presentation in the AI and ACC (Paulus et al. 2015). For stimuli with durations of about 3-to 5 seconds we think that the slope is a very good measure to capture arousal related pupil dynamics. For longer stimuli - when arousal related pupil dilation reaches a plateau – taking the differences between onset and a window at the end of a stimulus might be better suited as a summary measure (see e.g. (Krach et al. 2015)) which is however not the case in our study (see Figure R1).

Second, the reviewer addresses “information” response of the pupil in their comment. Information (or “surprise” in the context of the current experiment) peaks earlier and is a more temporally constrained response. For example, results from (Preuschoff 2011) indicate a peak increase for surprising events at around 1 second after stimulus onset with strong increase and decline shortly beforehand and after. The exact timing depends on the complexity of the depicted information and could also arise a little later in line with the suggestion of the reviewer. From this perspective, our measure would of course not be optimal for capturing the surprise response of the pupil, which requires a more circumscribed and earlier window or completely different approach (see below). However, since we do not compare valence and surprise dependent changes in the pupil and valence being our main target for testing modulations of physiological responses by valence dependent learning biases or affective experiences, we think it is reasonable to neglect the surprise response and capitalize on sustained arousal related in pupil diameter.

One could of course run more sophisticated analyses of the pupil response and analyse differential effects of PE surprise and PE valence. Applicable in our case, we could for example run repeated LMMs on the full pupil trace from the beginning of the stimulus onset until the stimulus end, testing the hypothesis that PE surprise effects would be stronger and more circumscribed earlier at around 1.5 seconds, while valence effects would only start to build up around this window and increase or persist until the end. However, we think this approach would distract from our main findings and we would like to keep the analyses as they currently are.

To briefly clarify our rationale, we have now referred to this prior work in the method section to provide a rationale for why choosing the slope as a summary measure for a trial is meaningful.

Lines 640-45: "To characterize the pupil dilation for each trial by a single value, we calculated a linear slope for each feedback phase of three seconds. Summarizing the pupil dynamics during a single trial with a linear slope is a robust and valid measure for an arousal related pupil response to stimuli of comparable lengths (Paulus et al. 2015; Krach et al. 2015; Laura Müller-Pinzler et al. 2015) building up after 1-2 seconds until reaching a plateau after more than approximately 6 seconds (Geuter et al. 2014; Bradley et al. 2008)".

(5) [related to point above] The shape of the average pupil diameter trace (Fig 2b) made interpreting the results a bit tricky for me. The authors report that "more negative PEs are associated with greater pupil slopes compared to positive PEs" (Figure 2b). That's true, but the authors interpret this as being consistent with the negative learning bias they find in the computational model. But since all slopes (for positive and negative PEs) are negative in their analysis, another way of saying this is that negative PEs are associated with flatter slopes compared to positive PEs (which have much steeper slopes). And one could conceivably think of ways that steeper slopes might indicate greater arousal (e.g., dropping back down to baseline from a higher peak). This is not to say that the authors interpretation is necessarily wrong, it just seems to me that you could interpret the direction of this result in various ways...

As prior research indicates (Geuter et al. 2014; Bradley et al. 2008) and we have demonstrated earlier (Paulus et al. 2015; Krach et al. 2015; Laura Müller-Pinzler et al. 2015), arousal related pupil responses manifest later and build up after 1-2 seconds after stimulus onset. They are rather low-frequency signals which can be sustained and further increase in the subsequent seconds of stimulus presentation (Geuter et al. 2014). For this reason, the suggested alternate interpretation of the findings in the sense of high arousal at the beginning of the feedback and thus more negative decline of the slope to

baseline cannot explain an arousal related response during the 3 seconds of feedback presentation. Any early arousal differences in this window must be related to events prior to the feedback presentation. This is exactly the reason for why we did not compare the Self and Other condition; the offset at the beginning of the trial is caused by the greater arousal from the prior estimation of the properties in the Self condition. Comparing the slopes between both condition would not be valid (see also Comment #6 by reviewer #1 and Figure R1) and results could not be interpreted. Since there are no significant offsets for the PEs of different valence, the comparisons of the respective slopes should be fair.

Comment #5: fMRI

Line 259-261: I do not agree that the absence of a significant difference between self and other suggests that there is a common process of error tracking independent of the agent (lack of an effect for a difference between the conditions is not evidence that there is no difference).

- We agree with the reviewer that absence of evidence for different neural activity is not evidence for the absence of this effect. We changed our wording accordingly:

Lines 278-280: "There was no significant difference between Self and Other ($p < .001$ uncorrected), indicating that there is no evidence for distinct neural processes of error tracking mapped between agents."

Comment #6: I found a lot of the results here a bit difficult to join up with the behavioural/computational results that had gone before it. I think this is in part because the fMRI model uses a conceptually different approach to the computational modelling. Specifically, the learning model effectively separates trials into positive and negative information trials and uses one of these two learning rate in the update rule (depending which of these two trial types applies). The fMRI model does not do this. It collapses all trials

together and uses a signed PE to identify which areas of the brain BOLD scales with. But it could be that different regions respond to positive and negative information (e.g., region X correlates with the magnitude of positive PEs and region Y the magnitude of negative PEs) which I don't think would be detected using this approach.

- We agree with the reviewer that this approach would be closer to our behavioral data analyses. We would like to explain our reasoning in a bit more detail:

1. Our first argument relates to increased power of the analyses when combining both conditions. We only have 20 Trials per condition in a rather complex task and combining both conditions does not only double the number of trials but also increases the variance for PE valence values by combining positive and negative PE trials.
2. As we already discussed in the paper, prediction errors in our learning task always involve different components, two of them are the valence of the PE and the surprise. When assessing only positive PEs, surprise and valence are completely correlated. Thus, the effect of the parametric weights for Self-Positive will always include both components, while the positive effect of the parametric weights for Self-Negative only shows the valence effect and the negative effect for PE surprise. In general, contrasting Self-Negative vs Self-Positive does not allow us to distinguish if we find a distinct response to PE valence or if the effect is due to PE surprise. Combining both conditions and including both predictors at the same time allows us to control for shared variance and assess only their unique effects.

Here, we nonetheless show the results for the contrast of the parametric weights as suggested separately for Self-Negative vs Self-Positive for both PE valence and PE surprise and parameter estimates for some peak regions to illustrate that the effects here are mainly also represented in our current analyses combining both conditions. Figure R6 shows the effect of PE valence for Self-Positive > Self-Negative. Self-Positive entails the PE valence and also the PE surprise effect and the negative PE Valence effect of Self-Negative also

entails the positive PE surprise effect. As PE surprise effects add up and PE valence are subtracted here, the effect shown is therefore comparable to the PE surprise effect in our original analysis as indicated by the parameter estimates for the mPFC. Figure R7 shows the effect of PE surprise for Self-Positive > Self-Negative. Self-Positive again entails PE valence besides the PE surprise effect and the negative PE surprise effect of Self-Negative also entails the positive effect of PE Valence. Here, PE valence effects add up and PE surprise are subtracted. Results are therefore comparable to the PE valence effect in our original analysis as indicated by the parameter estimates for the ventral striatum. The opposite comparisons (PE valence for Self-Negative > Self-Positive and PE surprise for Self-Negative > Self-Positive) did not yield any further effects (Figure R8). We believe, as we are unable to disentangle the effects of PE surprise and PE Valence with this approach, our current analyses are more straightforward and parsimonious although they do not fully match the analyses of the behavioral results.

Figure R6. PE valence for Self-Positive > Self-Negative. LEFT: The flexible factorial model includes parametric weights for PEs for Self-Positive (1), Self-Negative (2), Other-Positive (3), and Other-Negative (4). The contrast PE valence for Self-Positive > Self-Negative is depicted and parameter estimates are shown for the peak voxel in the mPFC. The Other conditions are of no interest here. Parametric weights for PEs are coded as unsigned values which explains the contrast weights in the Self-Negative condition (i.e. -1 codes more positive PEs). RIGHT: The right side shows the corresponding PE surprise effect in our current analyses (see also Figure 4).

Figure R7. PE surprise for Self-Positive > Self-Negative. LEFT: The flexible factorial model includes parametric weights for PEs for Self-Positive (1), Self-Negative (2), Other-Positive (3), and Other-Negative (4). The contrast PE surprise for Self-Positive > Self-Negative is depicted and parameter estimates are shown for the peak voxel in the ventral striatum. The Other conditions are of no interest here. Parametric weights for PEs are coded as unsigned values which explains the contrast weights in the Self-

Negative condition (i.e. -1 codes more positive PEs). RIGHT: The right side shows the corresponding PE valence effect in our current analyses (derived from Figure 4).

Figure R8. Results for the opposite comparisons. RIGHT: PE valence for Self-Negative > Self-Positive and LEFT: PE surprise for Self-Negative > Self-Positive. The flexible factorial model includes parametric weights for PEs for Self-Positive (1), Self-Negative (2), Other-Positive (3), and Other-Negative (4). The Other conditions are of no interest here. Parametric weights for PEs are coded as unsigned values which explains the contrast weights in the Self-Negative condition (i.e. -1 codes more positive PEs).

Comment #7: [sidenote: this approach also makes interpreting the later analysis - with predefined ROIs – tricky and I am not sure I agree with some of the interpretation as it is. E.g., the authors interpret (lines 288-292) that the more negative the learning bias is, the more neural activity responds to negative vs positive (see Figure 4d, left panel). But I don't think that is quite right. If I understood the analysis correctly, it would be more accurate to say that the more negative the learning bias is, the less sensitive neural activity is (i.e. the weaker the parametric effect is) in these regions to the valence of the PE (but it may be for instance that

at all levels of learning bias, the sign of the parametric regressor is positive, i.e. greater activity for positive PEs relative to negative PEs).

- We agree with the reviewer that the interpretation of the data depends on whether the parameter estimates of the parametric weights are always positive or distributed around zero. Our interpretation of increased responses in both directions, positive and negative, depending on the value of the covariates came from the main effect of PE valence and parameter estimates in our ROIs. For all ROIs individual parameter estimates are distributed around zero as depicted in the plot below, indicating that some participants show a stronger response with more positive PE valence while others show a stronger response with more negative PE valence. We added this Figure to the Supplementary Information (Figure S5). Therefore, when looking at the correlations it makes sense to say that the more negative the learning bias is, the more neural activity responds to negative vs positive and vice versa.

Figure S5: Parameter estimates for our predefined ROIs (amygdala, vAI, dAI, VTA/SN, and mPFC) were derived from all voxels within each ROI and averaged across all voxels. Left and right amygdala, vAI, and dAI were combined bilaterally for displaying purposes. Dots show the data for individual subjects, the cross marks the median and the line the mean. All regions show variance

between subjects in such a way that some individuals have positive values and other negative values indicating stronger activity scaling with more positive or more negative PEs, respectively.

Comment #8: The same interpretational issue applies to other analysis using individual difference measures (ratings, pupillometry).]

- As stated by the reviewer that the same reasoning applies for our pupil analyses. In order to support our interpretation, we plotted the data in Figure 2 b and 2 b. Figure 2 b shows three exemplary subjects one of which responds with greater pupil dilation to more negative PEs, another doesn't show differences in pupil dilation with respect to PEs, and the third shows increased pupil dilation scaling with more positive PEs. This already illustrates, similar to the fMRI data, that there is variance between subjects and some individuals respond more strongly to more negative PEs while others do the opposite. Figure 2 c specifically shows the associations with the three covariates for higher, average, and lower values of each covariate. We tried to frame our interpretation of the data as closely based on the data and these plots as possible (e.g. "These analyses demonstrated that individuals who experienced more embarrassment showed stronger pupil dilations scaling with more negative PEs, while pupil slopes did not correlate with PEs in individuals with lower embarrassment" as depicted in Figure 2 c, left).

Comment #9: To address some of this, I think it would be good if the authors modelled the fMRI data with a different GLM in which, instead of having signed PEs and absolute PEs as parametric regressors (pmods) in the same GLM (modulating the timepoint information is presented), they instead have only absolute PEs as pmods but split the time information is presented into two events "positive valence" events and "negative information" events. That way, it might be easier to identify (a) the different and overlapping brain regions for each type of valence, (b) contrast parametric effects of positive with negative, (c) relate activity to learning rates and subjective ratings.

- This comment relates to comment #6 above. There, we provided a more detailed answer. We explained our rationale behind our current analyses in more detail and also showed results for the GLM with separate conditions for positive and negative PEs as suggested by the reviewer. In addition, results for (a) the different and overlapping brain regions for each type of valence are now presented in the Supplementary Results section.

Reviewer #3 (Remarks to the Author):

Review of Neurocomputational mechanisms of affected beliefs by Müller-Pinzler et al.

Summary

In this study, the authors investigated the neurocomputational mechanisms underlying the interplay between beliefs formation about self-efficacy and affects. To that end, they analyzed the behavior, pupil data and BOLD responses of 39 subjects (and additional behavior of 30 other subjects) performing a task involving a series of cognitive estimations (e.g., estimating the weight of animals) and efficacy predictions. Feedback on efficacy was manipulated such that a low ability condition (negative feedback) and a high ability condition (positive feedback) were created. Subjects could perform the task themselves in one condition or predict the efficacy and observe the behavior of others in a second condition. Different learning models were fitted to the data to highlight learning mechanisms. Authors found that subjects tended to take more into account negative feedback about their self-efficacy when they performed the task themselves whereas they learnt in an unbiased manner when they observed others perform the task. They also found that this learning bias was related to experiences of pride and embarrassment, pupil size and neural activity.

The question investigated in the present study (i.e., what are the neurocomputational bases of "affected beliefs" formation?) is of prime interest, and numerous and various analyses have been used to answer that question. Overall, the analyses are clear and well written, and figures are precise. However, behavioral, neural and pupillometry hypotheses are not very well stated in the current version of the manuscript and the paper tends to look like a collection of analyses without clear justifications. I also have a series of major and minor comments, particularly on the behavioral and computational parts.

- We thank the reviewer for this remark. Reviewer #1 raised the same concerns about the clarity of our hypotheses and the presentation of the results. In line with both suggestions, we reduced the analyses in the main manuscript to the most necessary ones and we moved several analyses to the Supplement (e.g. average brain responses to feedback Self > Other) and removed some analyses entirely (e.g. the third GLM assessing neural activity in response to different

components of PEs). We also specifically stated our hypotheses in the beginning of each section (behavior, pupil, brain).

Major Comments

#1 LOOP task and predefined prediction errors

Comment #1a: In the task used in this study, feedback is manipulated such that a series of predefined prediction errors is presented to subjects, instead of a series of predefined outcomes (as in previous versions of the task). This feature implies that the more extreme subjects' expectations are, the even more extreme the feedback will be, and in a similar fashion, the more subjects update their expectations (high learning rate), the more extreme the feedback becomes. It would then be interesting to see how actual feedback differed between subjects. Was the feedback received by biased subjects more extreme (close to 0) in the low ability condition versus the high ability condition (close to 100)? Was the negative feedback closer to 0 for biased subjects versus unbiased subjects?

Comment #1b: It is of prime importance insofar as pride and embarrassment ratings could potentially be linked to the more extreme feedback received rather than to the higher learning rate for negative PE. Could the authors comment on this?

- We agree with the reviewer that there are dependencies of feedback, expectation, and learning rates due to the new and fMRI-adapted task design. By trying to achieve a uniform distribution of PEs to minimize systematic biases in the fMRI results, new problems unfortunately arise. The final feedback for the positive self-related condition is on average less close to 100 (mean=65.0, standard deviation=14.1) than the negative feedback is close to 0 (mean=27.5, standard deviation=14.5). By the fact that feedback also differs between individuals and learning biases arise from expectations, which in turn are related to feedback, there is also a relationship of feedback and learning biases. To show whether subjects' emotion ratings are related to feedback and whether the relationship between emotions and learning bias can be explained by different feedback values, we calculated partial correlations for emotion

ratings and the Valence Learning Bias, controlled for the mean feedback that subjects received before the emotion ratings. The partial correlations show that the relationship between emotion ratings and Valence Learning Bias does not significantly change and the overall pattern of associations remains similar (pride: $\rho(68)=.51$, $p<.001$; embarrassment: $\rho(68)=.20$, $p=.110$), although feedback is also correlated with emotion ratings ($\rho(68)=.28$, $p=.020$), as suspected.

See lines 218-221: "When controlling for differences in the feedback participants received before rating their affective experience correlations between emotions and Valence Learning Bias do not significantly change and the overall pattern of associations remains consistent."

#2 w parameter in the winning model

Comment #2a: In the winning model (M8), learning rates are reduced for the most extreme feedback. It is not clear how this parameter interacts with learning rates themselves. Simulations of the winning model could help understand how learning rates are impacted by different value of w. Also, it would be useful to see the fitted w parameters for each group of subjects (low vs. high bias) or potentially the learning rates from the Valence model without w (Model 5). Is w higher in the high bias group?

Comment #2b: As suggested in major comment #1, if more biased subjects obtained more extremely negative feedback, which are among those the most affected by w, it is not clear whether the effect of a higher learning rate for negative feedback is compensated or not by the effect of w.

- To answer the question, how w is related to learning rates and learning bias in the model we correlated all parameters within the winning model with each other (see Table S11). W is higher for individuals with a more positive learning bias and is also negatively correlated with the negative self-related learning rate. This answers the first part of the question, whether w is correlated with other parameters in the model. However, as we understand the reviewer's

comment, the more important aspect is whether w alters or compensates any effects in the model thus confounding the results. To answer this, as suggested, we also assessed parameter correlations for the Model 5 to assess whether w compensates or alters any effects that we would otherwise not find. The correlations between the same parameters from the different models are rather high (Valence Learning Bias: $\rho=.87$, $p<.001$, positive LR: $\rho=.82$, $p<.001$, negative LR: $\rho=.86$, $p<.001$) and we also find a similar negative Valence Learning Bias in both models (testing both Valence Learning Biases against zero: Model 8: mean=-.12, $t=-2.97$, $p=.004$; Model 5: mean=-.12, $t=-3.13$ $p=.003$) indicating that including w does not change the results or interpretation in a meaningful way. Although, w is correlated with learning bias and learning rates, parameter correlations between models indicate that there is no shift in learning rates or bias in such a way that individuals with a high bias according to Model 5 end up in “the low bias group” according to Model 8. We hope that this answers the reviewer’s question appropriately.

- ***Lines 186-188: "Exploratory analyses with learning rates from Model 5 showed that our results were unaffected by the w parameter modulating learning from more extreme feedback in the winning Model 8 (see Supplementary Results)."***
- ***Supplementary Results lines 134-141: "To assess whether introducing w as an additional parameter compared to our previous publications (Müller-Pinzler et al., 2019) we assessed correlations between the same parameters from the simple Valence Model 5 and our winning Model 8. Parameter correlations were rather high (Valence Learning Bias: $\rho=.87$, $p<.001$, $\alpha_{Self/PE+}$: $\rho=.82$, $p<.001$, $\alpha_{Self/PE-}$: $\rho=.86$, $p<.001$) and we also found a similar negative Valence Learning Bias in both models (testing both Valence Learning Biases against zero: Model 8: mean=-.12, $t=-2.97$, $p=.004$; Model 5: mean=-.12, $t=-3.13$ $p=.003$) indicating that including w does not change the results or interpretation in a meaningful way."***

Comment #3: Expectations initialization

The learning curves in figure 1a look extremely similar (self vs. other) with differences mainly in the initial predictions. In the task, initial expectations are set to 50% to compute manipulated feedback while in the models, they are set as free parameters. Could the authors explain the rationale behind this parametrization? Value initializations seem important when evaluating learning rate asymmetries and it would be interesting to check whether this difference in value initialization could explain in part the difference in learning rates.

- If we understand the question correctly, the reviewer asks whether the difference in learning bias between the Self and Other condition, i.e. the negative bias for Self and the absence of it for Other, is related to the fact that different initial values were estimated in the model between the conditions. Our rationale behind estimating the initial values instead of fixing them to a specific value like 50 % is twofold. First, in our previous studies, we tested several options for initial values and the models with free parameters won over the models with fixed parameters in model comparisons (L. Müller-Pinzler et al. 2019). Second, we believe that variance due to subject specific initial ratings/ beliefs that we would not capture with the initial values would end up in the learning rates and confound learning rates. We argue, thus, that in our case the opposite of what the reviewer suggests would be true: Subjects have more positive initial ratings/ beliefs for the Other as compared to the Self (see Table S8). Fixing the initial value to 50 % for the Other condition although participants start with their subjective beliefs at e.g. close to 60 % would falsely increase positive learning rates for Other and confound learning biases between Self and Other. To show that the difference in learning biases between Self and Other is independent of the initial values, we calculated a repeated measures ANOVA comparing the bias for Self and Other while controlling for the initial values for all four ability conditions. Results indicate that the differences between biases is still significant ($F(1,64)=8.64, p=.005$).

Affect measures

Comment #4a: Affect scores used to categorize subjects are computed as the average of only two measures per condition (self vs. other). Furthermore, the only difference between those measures is the condition subjects were in in the previous trial. Could authors explain why they think that this score represents uniquely affects for one of the two conditions (self or other)?

- This question goes in a similar direction as comments #1 and #3 from Reviewer 2. Like we replied above, we would like to respond at different levels. First, we do not assume that the emotion is a completely specific response to a single trial, since the emotion ratings also correlated strongly between the two Self conditions (embarrassment ($\rho=.76$) and pride ($\rho=.62$)), just as the mean ratings correlate strongly between Self and Other (embarrassment ($\rho=.85$) and pride ($\rho=.90$)). Rather, it is a more general emotional state triggered by the task. Given the theoretical basis of the emotions pride and embarrassment, which are intrinsically self-related emotions, we argue that the ratings are related to the self by encompassing self-related appraisals and activation of self-concepts. We also specifically asked how the person feels him or herself for each rating. As a result, the ratings in the Other condition should also relate to one's own feelings. However, there is also a specific component that goes into the emotion ratings, as we argued above. A repeated measures ANOVA comparing the ratings of the 4 conditions thus shows that the pride ratings differ between conditions (high ability vs low ability) and there is a greater increase in pride ratings for Self (for high ability vs low ability) than for Other, as one would also expect based on theory and other studies (Stolz et al. 2020). This, however, does not imply that emotion ratings in the Other condition are fundamentally different from those in the Self condition. A correlation with the learning bias also shows that the emotion ratings averaged across all conditions yield correlations in the same direction (embarrassment ($\rho=-.22$) and pride ($\rho=.60$)) as only for the Self ratings. However, since there is a specific component between conditions and the emotions are related to the self due to their theoretical basis, we still consider it reasonable to average only the

ratings from the Self condition for the analyses. We also believe that averaging across all four conditions would raise similar questions in the other direction. Overall, we do not want to argue for trial or condition specific effects in our correlational results but still believe that emotion ratings are related to the self.

Comment #4b: In the manuscript, it is repeatedly stated that emotional states shift preferences for information of positive or negative valence (e.g., lines 117-118). However, measures of emotional states are recorded after learning phases. It would be interesting to fit the model separately before and after these emotional states measures and look at the difference in learning rates at these different times. Could authors comment on this?

- We agree with the reviewer that additional analyses could further delineate these associations. However, given the current setup of our task, we think it is not feasible to precisely address this comment with the present data. We only have very few emotion ratings in the current version of the task and we also only have 20 trials per condition. We also think that far less than 20 trials per condition would not be ideal to model complex learning models with several free parameters. A more promising approach might be to increase the number of emotion ratings throughout the task and directly include emotion ratings in the modeling procedure. To do this, we would have to figure out how to increase the number of ratings while preventing problems with the cover story / credibility of the task. One way would be to skip the Other condition and focus on the Self - an approach we would like to try in future studies.

Minor Comments

Comment #5: Figure 2d and 2e would benefit from a harmonization of the Y-axes in the scatter plots and line plots respectively. The plots are difficult to compare because of their different Y-axis.

- In line with the reviewer's suggestion we adjusted all axes in Figure 2d and 2e accordingly.

Comment #6: It is not clear why 30 additional subjects have been added to the study and the difference in sample size between the behavioral / computational analyses and the pupillometry / neural analyses is not very well stated.

- Yes, we agree that the description of the sample size in the results section was suboptimal. We have now clarified the sample description. Also, in the Methods section, we have a more detailed description of the sample and the allocation of participants to sub-group analyses.
- ***Lines 141-146: "In the present experiment, n=39 subjects (26 females, aged 18-28 years; M=22.3; SD=2.65) completed the task in the MRI. Another n=30 subjects (24 females, aged 18-32 years; M=23.3; SD=3.97) completed the task outside the MRI as a behavioral study. During the MRI scanning, eye-tracking data was additionally obtained in all but three subjects (see Methods section for more details). To examine the formation of self-efficacy beliefs we used the "Learning Of Own Performance" (LOOP) task (L. Müller-Pinzler et al. 2019; Czekalla et al. 2020)."***
- ***Lines 468-478: "In the MRI 39 participants (26 females, aged 18-28 years; M=22.3; SD=2.65) completed the study. We initially recruited 48 participants, but had to exclude six participants who did not believe the cover story of the task and three participants who did not attentively complete the task until the end (e.g. participants reported that they were too tired or the ratings indicated that they stopped responding to the estimation task). During the MRI scanning, eye-tracking data was additionally obtained and could be analyzed in all but three subjects who had insufficient data quality (resulting in n=36 for pupil data analyses). We recruited an additional 30 participants (24 females, aged 18-32 years; M=23.3; SD=3.97), who completed the study as a behavioral study outside the MRI to increase the sample size for***

computational modeling results (resulting in an overall N=69 for behavioral data analyses)."

Comment #6: There's a repetition line 235/236:

234 It has been suggested that pupil dilation reflects differences not only between stimuli
235 but similarly between individual biases during decision making (see Figure 2d for
examples of individual differences (see Figure 2d for examples of individual differences)

- Thanks, we removed the repetition.

References

- Bradley, Margaret M., Laura Miccoli, Miguel a. Escrig, and Peter J. Lang. 2008. "The Pupil as a Measure of Emotional Arousal and Autonomic Activation." *Psychophysiology* 45 (4): 602–7.
- Bromberg-Martin, Ethan S., and Tali Sharot. 2020. "The Value of Beliefs." *Neuron* 106 (4): 561–65.
- Cecchi, Romane, Fabien Vinckier, Jiri Hammer, Petr Marusic, Anca Nica, Sylvain Rheims, Agnès Trebuchon, et al. 2022. "Intracerebral Mechanisms Explaining the Impact of Incidental Feedback on Mood State and Risky Choice." *ELife* 11 (July). <https://doi.org/10.7554/eLife.72440>.
- Charpentier, Caroline J., Ethan S. Bromberg-Martin, and Tali Sharot. 2018. "Valuation of Knowledge and Ignorance in Mesolimbic Reward Circuitry." *Proceedings of the National Academy of Sciences of the United States of America* 115 (31): E7255–64.
- Czekalla, Nora, Janine Baumann, David S. Stolz, Annalina V. Mayer, Johanna Voges, Lena Rademacher, Frieder M. Paulus, Söre Krach, and Laura Müller-Pinzler. 2020. "Self-Beneficial Belief Updating as a Coping Mechanism for Stress-Induced Negative Affect." *BioRxiv Preprint*.
- Einhäuser, Wolfgang. 2017. "The Pupil as Marker of Cognitive Processes." In *Computational and Cognitive Neuroscience of Vision*, edited by Q. Zhao. Singapore: Springer Berlin Heidelberg.
- Geuter, Stephan, Matthias Gamer, Selim Onat, and Christian Büchel. 2014. "Parametric Trial-by-Trial Prediction of Pain by Easily Available Physiological Measures." *Pain* 155 (5): 994–1001.
- Joshi, Siddhartha, Yin Li, Rishi M. Kalwani, and Joshua I. Gold. 2016. "Relationships between Pupil Diameter and Neuronal Activity in the Locus Coeruleus, Colliculi, and Cingulate Cortex." *Neuron* 89 (1): 221–34.
- Kelly, Clare, Roberto Toro, Adriana Di Martino, Christine L. Cox, Pierre Bellec, F. Xavier Castellanos, and Michael P. Milham. 2012. "A Convergent Functional Architecture of the Insula Emerges across Imaging Modalities." *NeuroImage* 61 (4): 1129–42.
- Krach, S., I. Kamp-Becker, W. Einhäuser, J. Sommer, S. Frösse, A. Jansen, L. Rademacher, L.

- Müller-Pinzler, V. Gazzola, and F. M. Paulus. 2015. "Evidence from Pupillometry and fMRI Indicates Reduced Neural Response during Vicarious Social Pain but Not Physical Pain in Autism." *Human Brain Mapping* 36 (11).
<https://doi.org/10.1002/hbm.22949>.
- Kunda, Z. 1990. "The Case for Motivated Reasoning." *Psychological Bulletin* 108 (3): 480–98.
- Kurth, Florian, Karl Zilles, Peter T. Fox, Angela R. Laird, and Simon B. Eickhoff. 2010. "A Link between the Systems: Functional Differentiation and Integration within the Human Insula Revealed by Meta-Analysis." *Brain Structure & Function* 214 (5–6): 519–34.
- Loewenstein, George. 2006. "Social Science. The Pleasures and Pains of Information." *Science*.
- Mesulam, M. M., and E. J. Mufson. 1982. "Insula of the Old World Monkey. I. Architectonics in the Insulo-Orbito-Temporal Component of the Paralimbic Brain." *The Journal of Comparative Neurology* 212 (1): 1–22.
- Müller-Pinzler, L., N. Czekalla, A. V. Mayer, D. S. Stolz, V. Gazzola, C. Keysers, F. M. Paulus, and S. Krach. 2019. "Negativity-Bias in Forming Beliefs about Own Abilities." *Scientific Reports* 9 (1). <https://doi.org/10.1038/s41598-019-50821-w>.
- Müller-Pinzler, Laura, Valeria Gazzola, Christian Keysers, Jens Sommer, Andreas Jansen, Stefan Frässle, Wolfgang Einhäuser, Frieder Michel Paulus, and Sören Krach. 2015. "Neural Pathways of Embarrassment and Their Modulation by Social Anxiety." *NeuroImage* 119 (October): 252–61.
- Paulus, F. M., S. Krach, M. Blanke, C. Roth, M. Belke, J. Sommer, L. Müller-Pinzler, et al. 2015. "Fronto-Insula Network Activity Explains Emotional Dysfunctions in Juvenile Myoclonic Epilepsy: Combined Evidence from Pupillometry and fMRI." *Cortex; a Journal Devoted to the Study of the Nervous System and Behavior* 65.
<https://doi.org/10.1016/j.cortex.2015.01.018>.
- Preuschhoff, Kerstin. 2011. "Pupil Dilation Signals Surprise: Evidence for Noradrenaline's Role in Decision Making." *Frontiers in Neuroscience* 5 (September): 1–12.
- Rouhani, Nina, and Yael Niv. 2021. "Signed and Unsigned Reward Prediction Errors Dynamically Enhance Learning and Memory." *eLife* 10 (March).
<https://doi.org/10.7554/eLife.61077>.
- Rutledge, Robb B., Archy O. de Berker, Svenja Espenhahn, Peter Dayan, and Raymond J. Dolan. 2016. "The Social Contingency of Momentary Subjective Well-Being." *Nature*

Communications 7 (May): 11825.

- Rutledge, Robb B., Nikolina Skandali, Peter Dayan, and Raymond J. Dolan. 2014. "A Computational and Neural Model of Momentary Subjective Well-Being." *Proceedings of the National Academy of Sciences of the United States of America* 111 (33): 12252–57.
- Scherer, Klaus R. 2005. "What Are Emotions? And How Can They Be Measured?" *Social Sciences Information. Information Sur Les Sciences Sociales* 44 (4): 695–729.
- Stolz, David S., Laura Müller-Pinzler, Sören Krach, and Frieder M. Paulus. 2020. "Internal Control Beliefs Shape Positive Affect and Associated Neural Dynamics during Outcome Valuation." *Nature Communications* 11 (1): 1230.
- Tangney, June Price, Jeff Stuewig, and Debra J. Mashek. 2007. "Moral Emotions and Moral Behavior." *Annual Review of Psychology* 58: 345–72.
- Vinckier, Fabien, Lionel Rigoux, Irma T. Kurniawan, Chen Hu, Sacha Bourgeois-Gironde, Jean Daunizeau, and Mathias Pessiglione. 2019. "Sour Grapes and Sweet Victories: How Actions Shape Preferences." *PLoS Computational Biology* 15 (1): e1006499.
- Vinckier, Fabien, Lionel Rigoux, Delphine Oudiette, and Mathias Pessiglione. 2018. "Neuro-Computational Account of How Mood Fluctuations Arise and Affect Decision Making." *Nature Communications* 9 (1). <https://doi.org/10.1038/s41467-018-03774-z>.
- Zhang, Lei, Lukas Lengersdorff, Nace Mikus, Jan Gläscher, and Claus Lamm. 2020. "Using Reinforcement Learning Models in Social Neuroscience: Frameworks, Pitfalls and Suggestions of Best Practices." *Social Cognitive and Affective Neuroscience* 15 (6): 695–707.

REVIEWERS' COMMENTS:

Reviewer #1 (Remarks to the Author):

The authors have carefully addressed my concerns and provided additional information which increased the clarity of the applied design and analysis.

Reviewer #2 (Remarks to the Author):

this is a very robust rebuttal which the authors have clearly put a lot of careful time and effort into - I am happy to recommend this paper for acceptance

Reviewer #3 (Remarks to the Author):

Review of Neurocomputational mechanisms of affected beliefs by Müller-Pinzler et al. - round II

I thank the authors for their detailed and clear answers to my concerns. They provided a pertinent set of explanations, statistical tests, and clarifications of the methods together with an increased clarity of the manuscript in general, that improve substantially an already nice paper.

Minor comments

Line 205 - typo at the end of the sentence

203 We hypothesized that self-efficacy belief formation is associated with the affective
204 experience. In line with 2 we expected that individuals with more negative affective experience
205 would update their self-efficacy beliefs in a more on negative way.

Line 251 - a parenthesis is missing

249 It has been suggested that pupil dilation reflects differences not only between stimuli
250 but similarly between individual biases during decision making (see Figure 2b for examples
251 of individual differences.

Line 389 - typo at the beginning of the sentence

389 One the neural level, the anterior insula (AI), specifically, has been suggested to
390 function as an integrative hub for motivated cognition and emotional behavior

Reviewer #3 (Remarks to the Author):

Minor comments

Line 205 - typo at the end of the sentence

203 We hypothesized that self-efficacy belief formation is associated with the affective 204
experience. In line with 2 we expected that individuals with more negative affective
experience

205 would update their self-efficacy beliefs in a more on negative way.

- Thanks. We removed the word “on”.

Line 251 - a parenthesis is missing

249 It has been suggested that pupil dilation reflects differences not only between stimuli
250 but similarly between individual biases during decision making (see Figure 2b for
examples

251 of individual differences.

- We added a parenthesis.

Line 389 - typo at the beginning of the sentence

389 One the neural level, the anterior insula (AI), specifically, has been suggested to
390 function as an integrative hub for motivated cognition and emotional behavior

- We changed the word “one” to “on”.